# How soap bubbles freeze

S. Farzad Ahmadi [1], Saurabh Nath[1,2], Christian M. Kingett[1], Pengtao Yue[3] & Jonathan B. Boreyko [1,4]

Droplets or puddles tend to freeze from the propagation of a single freeze front. In contrast, videographers have shown that as soap bubbles freeze, a plethora of growing ice crystals can swirl around in a beautiful effect visually reminiscent of a snow globe. However, the underlying physics of how bubbles freeze has not been studied. Here, we characterize the physics of soap bubbles freezing on an icy substrate and reveal two distinct modes of freezing. The first mode, occurring for isothermally supercooled bubbles, generates a strong Marangoni flow that entrains ice crystals to produce the aforementioned snow globe effect. The second mode occurs when using a cold stage in a warm ambient, resulting in a bottom-up freeze front that eventually halts due to poor conduction along the bubble. Blending experiments, scaling analysis, and numerical methods, the dynamics of the freeze fronts and Marangoni flows are characterized.

[1] Department of Biomedical Engineering and Mechanics, Virginia Tech, 495 Old Turner Street, 222 Norris Hall, Blacksburg, VA 24061, USA. [2] Physique et Mécanique des Milieux Hétérogènes, UMR 7636 du CNRS, ESPCI, PSL Research University, 75005 Paris, France. [3] Department of Mathematics, Virginia Tech, 225 Stanger Street, 460 McBryde Hall, Blacksburg, VA 24061, USA. [4] Department of Mechanical Engineering, Virginia Tech, Blacksburg, VA 24061, USA. Correspondence and requests for materials should be addressed to J.B.B. (email: boreyko@vt.edu)

Soap bubbles and films have been a source of intrigue for millennia. Their influence can be traced in a historical arc that includes Babylonian divination rituals (lecanomancy) to Impressionist paintings to the works of physicists such as Newton, Plateau, and de Gennes[1–4]. Building on this foundation, in modern times, the behavior of bubbles from their birth[5,6], wetting[7], drainage and evaporation[8,9], to their fatal bursting[10,11] has been comprehensively studied. This has culminated in the practical use of bubbles for a myriad of applications such as energy harvesting[12], drug delivery[13], and cleaning devices[14–16].

Despite this prolonged attention lavished on bubbles, there exists only a brief scientific report of freezing. In 1949, Shaefer observed bubbles freezing atop Mt. Washington and commented on the number and shape of ice crystals contained therein[17]. In the field of visual arts, on the other hand, there is an emerging trend of photographers capturing beautiful videos of the complex freezing dynamics of bubbles deposited on snow (e.g., see https://www.youtube.com/watch?v=H7pqoCJQp2I). These dynamics are nontrivial owing to the unique geometry of a bubble: unlike droplets, puddles, or surface-bound liquid films, bubbles do not have a thermally conductive bulk volume. Therefore, the extensive studies of how droplets[18–25] or films[26–28] freeze cannot capture the physics of bubble freezing.

Inspired by these informal observations of freezing bubbles, here we characterize the heat transfer phenomena governing the dynamics of freezing bubbles over a wide range of conditions. Two different types of freezing dynamics were observed, depending on the experimental conditions. For bubbles freezing in an isothermal environment, the bottom-up freeze front produces a Marangoni flow that detaches ice crystals, resulting in accelerated freezing from multiple fronts growing in tandem. Conversely, when bubbles are deposited on a chilled icy substrate in room-temperature conditions, the bottom-up freeze front comes to a halt midway up the bubble due to poor conduction.

## Results

**Experimental setup.** Two separate sets of experiments were performed: isothermal experiments, where a walk-in freezer set both the ambient ($T_\infty$) and substrate temperature ($T_w$) far beneath the melting point ($T_m$): $T_\infty \approx T_w = -18 \pm 2\,°C < T_m$ (Fig. 1a), and room-temperature experiments, where the ambient was warmer than the melting point and freezing was accomplished with a chilled substrate: $T_\infty \approx 25\,°C > T_m > T_w$ (Fig. 1b). In both scenarios, the bubble was deposited on an icy substrate, such that the freezing process could begin immediately. For isothermal experiments, soap bubbles enclosing air volumes of $\Omega = 500\,\mu L$ or $10\,mL$ were deposited onto an ice disk. For the room-temperature experiments, $\Omega = 5\,\mu L$ or $500\,\mu L$ and the substrate was cooled anywhere from $T_w = -10\,°C$ to $-40\,°C$ and allowed to frost over. These two different conditions for the ambient produced two distinct modes of bubble freezing.

For bubbles in both sets of experiments, we used a glycerol–water soap solution exhibiting a freezing point of $T_m \approx -6.5\,°C$ (see the "Methods" section)[29]. When a calm bubble is punctured, a hole opens and grows due to the unstable surface tension forces at its rim. Using the Dupré–Taylor–Culick law, the initial film thickness of a liquid bubble ($e_0$) can be determined from the hole-opening velocity by balancing surface tension and inertia[30,31]:

$$v_b = \sqrt{\frac{2\gamma}{\rho e_0}}, \tag{1}$$

where $\gamma = 24.2\,mN\,m^{-1}$ is the solution's surface tension measured by using the pendant drop method and waiting until the surfactant had reached a steady-state packing density at the

free interface (Supplementary Fig. 1). Using high-speed imaging, bursting velocities of $v_b \approx 3.2\,m\,s^{-1}$, $v_b \approx 4.1\,m\,s^{-1}$, and $v_b \approx 5.3\,m\,s^{-1}$ were observed for $\Omega = 5\,\mu L$, $\Omega = 500\,\mu L$, and $\Omega = 10\,mL$ bubbles, respectively (Supplementary Fig. 2). Given that the soap solution is 80% water, we approximate the density (and all other thermophysical properties besides $T_m$) as that of pure water: $\rho \approx 1000\,kg\,m^{-3}$. From Eq. 1, we obtain $e_0 \approx 4.7\,\mu m$ for $\Omega = 5\,\mu L$, $e_0 \approx 2.7\,\mu m$ for $\Omega = 500\,\mu L$, and $e_0 \approx 1.7\,\mu m$ for $\Omega = 10\,mL$.

**Bubble freezing under isothermal conditions.** Figure 2 shows the remarkable multistep freezing process that occurs under isothermal conditions, over a timescale of $\mathcal{O}(10\,s)$ for centimetric bubbles. The initial mode of freezing was a bottom-up freeze front, analogous to the bottom-up freezing of sessile droplets but more dendritic in appearance. Immediately upon contact with the icy substrate, the bubble exhibited an upward flow of velocity $v \approx 10\,mm\,s^{-1}$ emanating from the early freeze front (Fig. 2a, b, and Supplementary Figs. 3 and 4). Within milliseconds, this flow destabilized into plumes with a radius $R_p \sim 1\,mm$. After a typical time of $\mathcal{O}(1\,s)$, ice crystals of about $100\,\mu m$ in diameter suddenly became visible and were entrained in the upward fluid flow. Hundreds of these entrained ice crystals swirled around the bubble in a dramatic fashion, which we term the "Snow-Globe Effect." After a few seconds, the plumes dissipated and died out as the ice crystals grew larger in size. Finally, the bubble completely froze over within $\mathcal{O}(10\,s)$, not only from the bottom-up freeze front, but also especially from the floating ice crystals as they grew and interlocked together.

Possible mechanisms: The mechanism for the vertical plumes that emerge when the bubble contacts the icy surface is far from obvious. At least four types of flows can be envisaged: Marangoni flow due to nonuniform surfactant concentrations, flow due to nonuniform surface curvature (marginal regeneration), buoyant flow due to thermal effects, or a thermal Marangoni flow induced by the latent heat of fusion. The first two flow mechanisms do not require a temperature gradient and can therefore be evaluated using bubbles deposited on a dry substrate at room-temperature conditions (Fig. 3a). The distribution of surfactant along the interface can be isolated by considering a bulk pendant drop suspended in air, where about 40 min was required to achieve a steady-state surface tension (Supplementary Fig. 1). However, recall that our soap bubbles are only about $1\,\mu m$ in thickness. For such thin films, having a large surface-area-to-volume ratio, the diffusion of surfactant to the free surface should be much faster[32]. For example, the ratio of diffusive timescales between the pendant drop and bubble is $R_{drop}^2/e_0^2 \sim 10^6$, such that a steady-state surface tension should be achieved within about 1 ms for the bubble. In contrast, we observed that room-temperature bubbles on dry substrates generated plumes over a very long timescale of $\mathcal{O}(10\,min)$ (Supplementary Fig. 5), ruling out asymmetric surfactant concentrations as a likely mechanism. Besides, the surface tension measured with the pendant drop was only changing temporally, not spatially, as the measured curvature indicated a single value of surface tension for any given time (Supplementary Fig. 1).

The second mechanism of marginal regeneration, as first proposed by Mysels, Shinoda, and Frankel[33], is purely geometric and owes its origin to the liquid meniscus at the foot of the bubble. The Laplace pressure difference between the bubble and the meniscus generates plumes that are long-lived[10,34]. This agrees with our control case of room-temperature bubbles, where the plumes were maintained for most of the lifetime of the bubble (Fig. 3a and Supplementary Fig. 5a). While we can therefore attribute plumes in the room-temperature bubbles to marginal regeneration, this does not necessarily explain the plumes in the

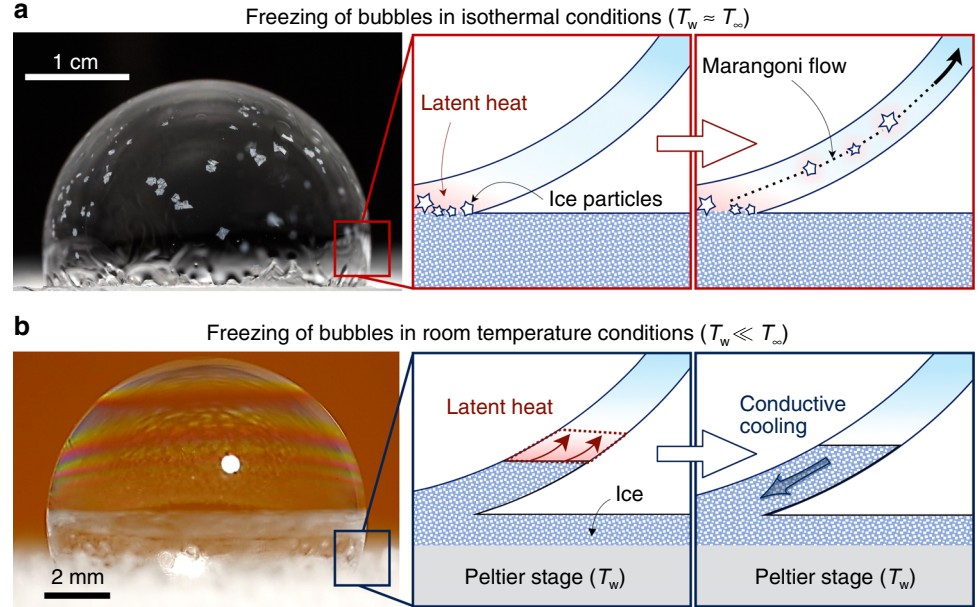

**Fig. 1** The dynamics of freezing bubbles under various ambient conditions. **a** For bubbles deposited on an icy substrate contained within an isothermal freezer, the freeze front induced local heating at the bottom of the bubble. This resulted in a Marangoni flow strong enough to detach and entrain growing ice crystals, such that the bubble froze from multiple fronts. **b** For bubbles deposited on a chilled, icy substrate in a room-temperature environment, the freeze front grew bottom-up in a uniform fashion before stopping entirely at a critical height. Latent heat generated at the growing freeze front had to be continually dumped into the substrate via inefficient conduction across the frozen portion of the bubble

freezing bubbles. For the case of freezing, the liquid meniscus will be solidified by the bottom-up freeze front within about 0.1 s (Supplementary Fig. 4d), which would halt the marginal regeneration. Considering that the plumes were observed to persist for several seconds for the freezing bubbles, we should instead consider the thermal mechanisms of buoyancy or Marangoni flow due to the latent heat released from the freeze front.

A previous work has shown that buoyant thermal plumes can be generated in vertical soap films, where the plumes were primarily inertial[35]. Inertia is negligible in our system, as the Reynolds number is $\mathrm{Re} = \rho V^2/(\eta V/e_0) = \rho V e_0/\eta$, where $\eta \approx 2 \times 10^{-3}$ Pa s is the viscosity of water[36] at $-6.5\,^\circ$C. For typical values of $V \sim 10$ mm s$^{-1}$, as measured by observing the initial speed of a rising plume (Fig. 2d), we get Re~0.01. A buoyant flow in our soap bubbles would therefore have to balance a gradient in pressure, $\Delta \rho g$, with the gradient in viscous stress, $\eta V/e_0^2$. For a typical value of $\Delta \rho \sim 1$ kg m$^{-3}$, this leads to buoyant flows of speed $V_\mathrm{B} \sim \Delta \rho g e_0^2/\eta \sim 10$ nm s$^{-1}$. This is in contrast to Fig. 2d, where the speed is not constant over time and is about six orders of magnitude faster ($V \sim 10$ mm s$^{-1}$).

This leaves us with the final possibility of a Marangoni flow induced by the latent heat released from freezing. We will refer to this process as "Marangoni freezing." The freezing-induced heating engenders a gradient in surface tension, $\Delta \gamma/\delta$, where $\delta$ is the length scale of the temperature gradient driving the flow. This must be balanced by viscous stress, $\eta V_\mathrm{M}/(b + e_0/2)$, where $V_\mathrm{M}$ is the Marangoni velocity, $b$ is the slip length of the Poiseuille flow along the bubble's film (Supplementary Fig. 6), and the velocity profile was approximated as a constant slope. For our system, $b = \sqrt{\eta R/(\rho g t_\mathrm{d})} \sim 1$ μm[8], $t_\mathrm{d} \sim 10^3$ s being the drainage timescale of a centimetric bubble, which was experimentally observed (Supplementary Fig. 5a). Therefore $(b + e_0/2) \sim e_0$, resulting in a simplified viscous stress of $\eta \dot{\delta}/e_0$, where $\dot{\delta} = \mathrm{d}\delta/\mathrm{d}t = V_\mathrm{M}$ represents the speed of a plume. Relating

the surface tension stress and viscous stress and solving for $\delta$:

$$\delta \sim \sqrt{\frac{2\Delta \gamma e_0}{\eta}}\, t^{1/2}. \qquad (2)$$

Note that $\Delta \gamma \approx 2$ mN m$^{-1}$ for $\Delta T = T_\mathrm{m} - T_\mathrm{l} \approx 13.5\,^\circ$C corresponding to our degree of supercooling (Fig. 2c)[37]. When comparing Eq. 2 to experiments, the trajectories of thermal plumes were tracked for $\Omega = 10$-mL bubbles (Fig. 2d). The measurements of $\delta$ are in good agreement with 1/2-law with a numerical pre-factor of 1.6, confirming the Marangoni freezing mechanism for flow in the freezing bubbles. Finally, the underlying physics for the resulting wavelength and plume radius ($R_\mathrm{P} \sim 1$ mm) are nontrivial and beyond the scope of this research, as has been noted before in the phenomenon of Marangoni bursting[38].

Marangoni freezing and the "Snow-Globe Effect": Marangoni freezing occurs when Marangoni flows can be generated by freezing-induced heating at the contact line, and these flows dominate over any other possible flow. The two temperature requirements for Marangoni freezing include the condition for freezing: $T_\mathrm{w} < T_\mathrm{m}$, and the condition for vertically upward Marangoni flow: $\mathrm{d}T/\mathrm{d}z < 0$, such that $\mathrm{d}\gamma/\mathrm{d}z > 0$[37]. For the Marangoni flow to dominate, it must be at least as fast as the rate of thermal diffusion, $V_\mathrm{T} \sim \alpha_\mathrm{l}/\delta$, where $\alpha_\mathrm{l} = 0.13 \times 10^{-6}$ m$^2$ s$^{-1}$ is the thermal diffusivity of the liquid solution. This first flow criterion is stated in terms of the ratio of the Marangoni velocity, $V_\mathrm{M} \sim \Delta \gamma e_0/\eta \delta$ (from Eq. 2), and $V_\mathrm{T}$. This ratio is called the Marangoni number, $\mathrm{Ma} = \Delta \gamma e_0/\eta \alpha_\mathrm{l}$, which should be greater than or equal to 1. The second flow criterion is that the velocity due to thermal buoyancy, $V_\mathrm{B} \sim \Delta \rho g e_0^2/\eta$, must be negligible compared with $V_\mathrm{T}$, resulting in a small Rayleigh number: $\mathrm{Ra} \sim \rho g e_0^2 \delta/\alpha_\mathrm{l} \eta \ll 1$. The above arguments can be succinctly summarized as $T_\mathrm{t} \lesssim T_\mathrm{w} < T_\mathrm{m}$, $\mathrm{Ma} \gtrsim 1$, and $\mathrm{Ra} \ll 1$, where $T_\mathrm{t}$ is the temperature of the top of the bubble. For the bubbles freezing in the walk-in freezer, all of the conditions were satisfied as

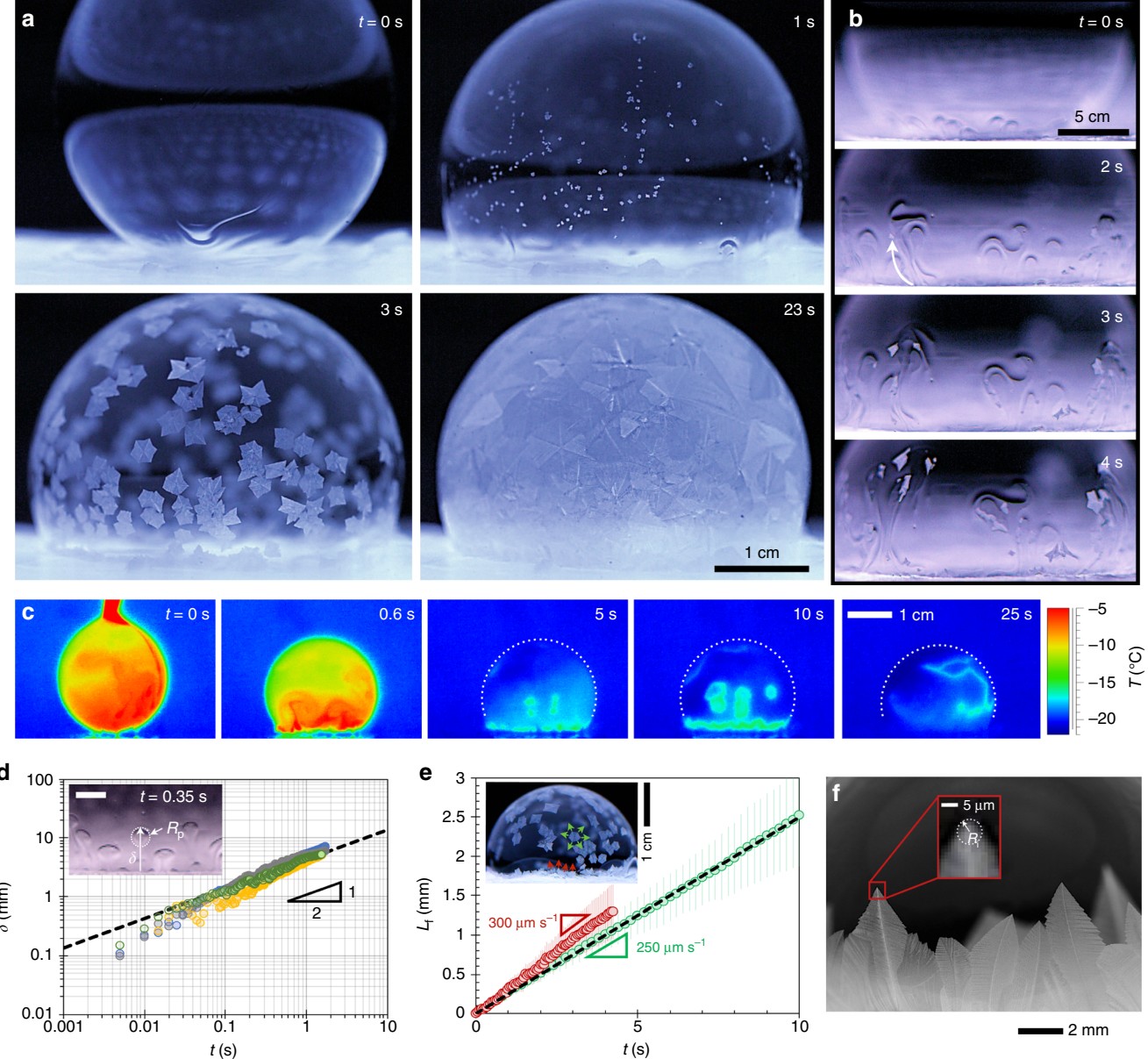

**Fig. 2** Freezing bubbles under chilled and isothermal conditions. **a** Freezing of a 10-mL bubble deposited on an ice disk (Supplementary Fig. 3) in a walk-in freezer chilled to $T_\infty = -18.5 \pm 0.5\,°C$ with RH = 60 ± 5%. **b** The freeze front induced a Marangoni flow, which detached and entrained some of the growing ice particles. **c** Time-lapse thermographic images, where dotted arcs clarify the bubble–air interface. The liquid portions of the bubble assumed the freezer's temperature shortly after deposition, while the freeze fronts were warmer (i.e., near the melting temperature) due to the release of latent heat. The emissivity coefficient of ice was calibrated at $\varepsilon = 0.98$. Time zero corresponds to the bubble's first contact with the icy substrate, where the top of the bubble is still adhered to the pipette (first frames of **a**–**c**). **d** Displacement ($\delta$) of four thermal plumes (different colors) was measured over time when $T_w \approx T_\infty = -19.6\,°C$. The inset shows that the radius of plumes was of order $R_p \sim 1$ mm. The scale bar represents 2 mm. **e** Growth rate of freeze fronts coming from the substrate (red data points, $v_i \approx 300\,\mu m\,s^{-1}$) or from ice crystals suspended in the liquid film (green, $v_i \approx 250\,\mu m\,s^{-1}$) when $T_\infty = -18.4 \pm 1.7\,°C$. Error bars represent a standard deviation from an average of three trials. **f** The ice radius, $R_i$, was estimated from the tip of ice crystals growing from the substrate

$T_t \approx T_w \approx -20\,°C < T_m = -6.5\,°C$, Ma ≈ 13, and Ra~$10^{-5}$. Besides generating the thermal plumes, Marangoni freezing can also be responsible for what we call the "Snow-Globe Effect," as will now be discussed.

Owing to the high Ma number, we propose that Marangoni flows shear off and entrain ice dendrites forming at the bottom-up freeze front. While we do not have any direct evidence of a flow shearing off an ice dendrite, as they are too small to be visible at the point of detachment, there are two strong justifications for

this claim. First, it is highly unlikely that hundreds of homogeneous nucleation events would suddenly occur within the liquid film away from the freeze front, especially considering that the freezer temperature is too warm to promote homogeneous nucleation. This was confirmed by depositing a bubble on a dry silicon wafer (still in the walk-in freezer), where no freezing/nucleation events were observed even after 30 min (Supplementary Fig. 3c). Second, whenever the suspended ice crystals first appeared (i.e., grew to a micrometric size), it was

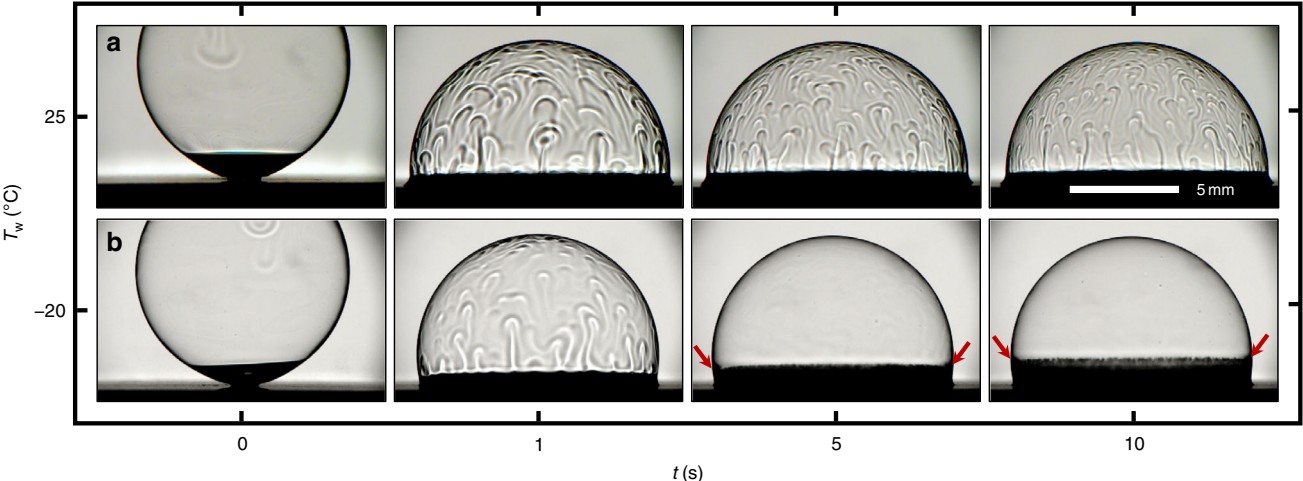

**Fig. 3** Contrasting mechanisms for plumes in nonfreezing vs. freezing bubbles. **a** For a bubble deposited on a dry, room-temperature substrate, plumes were continually generated through the ~10-min lifetime of the bubble due to marginal regeneration. **b** For a bubble deposited on an icy cold stage ($T_w = -20\,°C$), the bottom-up freeze front (red arrows) suppressed marginal regeneration but enabled a brief (~1 s) flow due to Marangoni freezing. In either case here, the ambient conditions were $T_\infty \approx 25\,°C$ with a relative humidity of RH $\approx 19\%$

always during the Marangoni flow. Indeed, the growth of the entrained ice crystals was often highly asymmetric due to the flow, as seen in Fig. 2b. The Marangoni flow must therefore be detaching invisibly small (i.e., nanoscale) ice particles from the bottom-up freeze front and advecting them upward. After about 1 s of Marangoni freezing, hundreds of microscopic ice particles were suspended and growing within the film, working in tandem to heat the surrounding liquid. At this point, the gradient in temperature and surface tension is happening in a myriad of locations and directions, as opposed to the original case of a fully out-of-plane gradient extending from the bottom-up freeze front. Thus, the "Snow-Globe Effect" annihilates the very Marangoni flow that created it in the first place.

This ice detachment can be tentatively modeled by balancing the inertia of the thermal plume ($F_i$) with the pull-off force required to crack an ice dendrite free of its icy substrate ($F_{crack}$). For a dendritic contact area of $\pi l^2$, the pull-off force can be determined using the Griffith condition for crack initiation[39]

$$F_{crack} = \pi l^2 \sqrt{\frac{8E^* w_{ad}}{\pi l}}, \qquad (3)$$

where $w_{ad}$ is the work of adhesion and $E^* = E_i/(1 - \nu_i^2)$. Here, $E_i = 8.7 \times 10^9\,Pa$ and $\nu_i = 0.31$, are the Young's modulus and Poisson's ratio corresponding to ice, respectively[40]. The work of adhesion can be quantified as $w_{ad} \approx \gamma_{i,l}$ where $\gamma_{i,l}$ is the interfacial energy of ice with respect to liquid determined from Young's relation, $\gamma_{i,l} = \gamma_{i,v} - \gamma\cos\theta$. Here, $\gamma_{i,v} \approx 0.1\,J\,m^{-2}$ is the interfacial energy of ice with respect to vapor[41], $\gamma \approx 0.02\,J\,m^{-2}$ (Supplementary Fig. 1), and $\theta \approx 0°$ is the intrinsic contact angle of the liquid solution on ice. Therefore, the work of adhesion between an ice dendrite and the icy substrate is $w_{ad} \approx 0.08\,J\,m^{-2}$. The inertia of a thermal plume is $F_i \sim (\rho \pi R_p^2)\nu_{M,0}^2$, where $\nu_{M,0}$ is the Marangoni velocity at the very early time limit. Experimentally, we find $\nu_{M,0} \sim 10\,mm\,s^{-1}$ by taking the derivative from the $\delta - t$ plot (Fig. 2d) at $t < 10\,ms$. Balancing the pull-off force and inertia, $F_i \sim F_{crack}$, predicts that a dendrite must be smaller than $l \lesssim 10\,nm$ for detachment. As shown in Fig. 2a, b, entrained ice particles grow to ~100 μm in size after $\approx 1\,s$ of bubble deposition on the icy substrate. This is consistent with the measured growth rate of ice of $\nu_i \sim 100\,\mu m\,s^{-1}$ (Fig. 2e), indicating that ice particles do indeed detach from the freeze front at a nanoscale size.

The growth rate of the ice front can be modeled by using the well-known two-phase Stefan problem, where a semi-infinite ($0 < y < \infty$) supercooled liquid with temperature $T_l < T_m$ is exposed to a temperature $T_c$ at its boundary ($y = 0$) at time zero. The tip velocity is then given by

$$\nu_i = 2\lambda^2 \alpha_i/R_i, \qquad (4)$$

where $\lambda \approx 0.03$ was obtained from the root of a transcendental equation (Eq. 5), $\alpha_i = 1.15 \times 10^{-6}\,m^2\,s^{-1}$ is the thermal diffusivity of ice, and $R_i$ is the tip radius of the ice layer[27]. For our case of $T_c = T_m$, the transcendental equation can be expressed as

$$\frac{St_l}{\nu\lambda\,erfc(\lambda\nu)\exp((\lambda\nu)^2)} = \sqrt{\pi}, \qquad (5)$$

where $\nu = \sqrt{\alpha_i/\alpha_l}$ is the ratio of the thermal diffusivity of ice to the water, $St_l = (c_w(T_m - T_l))/L$, $c_w = 4.2 \times 10^3\,J\,Kg^{-1}K^{-1}$ is the specific heat capacity of water, and $L = 334 \times 10^3\,J\,kg^{-1}$ is the latent heat of fusion. The tip diameter was crudely measured as $2R_i \approx 1.28\,\mu m$ (Fig. 2f), consistent with a previous report[27] and the need to be contained within the film ($e_0 \approx 1.7\,\mu m$, see Supplementary Fig. 2a). Plugging this value of $R_i$ into Eq. 4 gives a theoretical tip velocity of $\nu_i \approx 250\,\mu m\,s^{-1}$ (dotted line in Fig. 2e), in agreement with experimental growth measurements of $\nu_i \approx 300\,\mu m\,s^{-1}$ for the bottom-up freeze front and $\nu_i \approx 250\,\mu m\,s^{-1}$ for the crystals suspended within the bubble film.

**Bubble freezing under room-temperature conditions.** In a second set of experiments, bubbles were deposited on a cold stage set to temperature $T_w$ within a room-temperature ambient environment (Fig. 4a). Under these conditions, freezing progressed in four stages: Marangoni freezing, partially frozen equilibrium, marginal regeneration, and collapse.

Marangoni freezing: Analogous to the walk-in freezer experiments, even in room-temperature experiments, we see that mere milliseconds after deposition of bubbles on a chilled icy substrate, there is a burst of Marangoni plumes moving upward from the contact line. The difference between the two experiments is the initial imposed positive temperature gradient across the bubble in room-temperature experiments ($T_t > T_w$). This, however, has no effect on the initial stages of freezing, as the latent heat released in these experiments to create a local

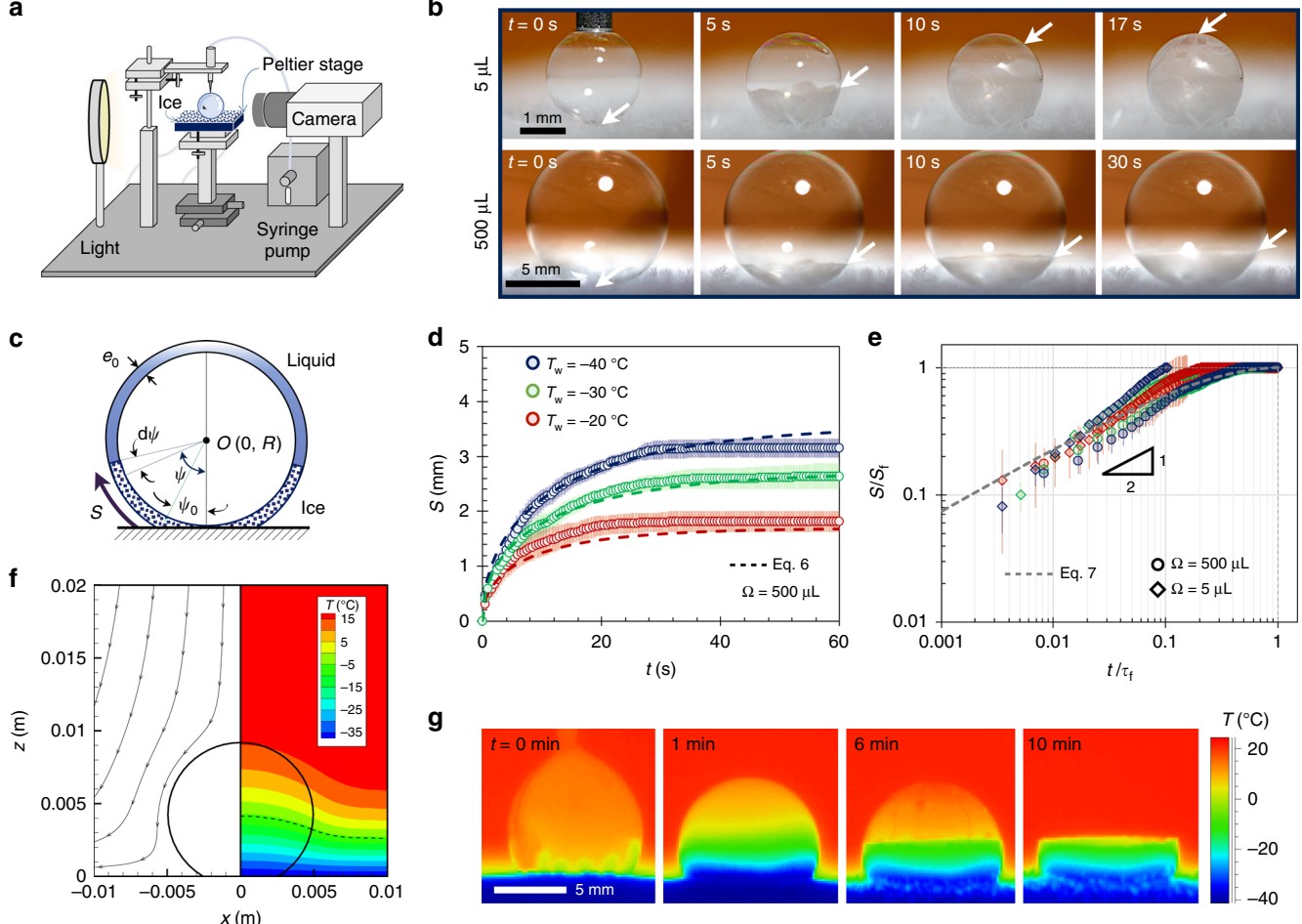

**Fig. 4** Freezing bubbles on a cold stage in a room-temperature environment. **a** Schematic of the experimental setup, where a bubble of controlled volume is deposited on a frosted Peltier substrate using a syringe pump. **b** Side-view imaging of freezing bubbles, of volume $\Omega = 5\ \mu L$ and $\Omega = 500\ \mu L$ on a surface chilled at $T_w = -40 \pm 1\ °C$. Arrows indicate the evolving location of the freeze front and time zero is when the bubble is first deposited. **c** Schematic showing the terms used in Eq. 6. **d** The frozen arc length of $\Omega = 500\ \mu L$ bubbles over time. Data points (circles) represent experimental data, with error bars of one deviation across three trials, while dashed lines represent Eq. 6. **e** For both $\Omega = 5\ \mu L$ (diamonds) and $\Omega = 500\ \mu L$ (circles), plotting the frozen arc length normalized by the final arc length ($S/S_f$) against a nondimensional timescale ($t/\tau_f$) collapsed all of the data onto a universal curve (Eq. 7). The average ambient conditions across all experiments were $T_\infty = 23.4 \pm 1.2\ °C$ and a relative humidity of RH $= 42 \pm 14\%$. **f** Simulation of the air temperature profile shows that near the center of the Peltier, the bubble somewhat disrupted the natural convection, resulting in greater slopes for the iso-temperature lines (see Supplementary Fig. 8). The dashed line corresponds to $T = -6\ °C$. **g** Thermographic images of a $\Omega = 500$-$\mu L$ bubble deposited on a frosted substrate of temperature $T_w = -40 \pm 1\ °C$ at a room temperature with $T_\infty = 23.3 \pm 1\ °C$ and RH $= 23 \pm 1.5\%$

region of $dT/dz < 0$. However, the negative temperature gradient is only a transient effect, such that the plumes would die out within about 5 s. Perhaps, due to the transient nature of Marangoni freezing for the case of a room-temperature ambient, the "Snow-Globe Effect" was not observed. For the larger $\Omega = 500$-$\mu L$ bubble, it is also possible that nanoscale ice particles could be entrained in the Marangoni flow, but quickly melted by the warm top of the bubble ($T_t > T_m$) before growing to a microscale size.

Partially frozen equilibrium: A bottom-up freeze front progressed up a bubble at an initial speed of $v \sim 0.1\ mm\ s^{-1}$ (Fig. 4b–e); this is similar to equivalent velocity of solidification fronts in water droplets[42]. However, unlike droplets, the freeze front of a bubble came to a complete stop after $\tau_f \sim \mathcal{O}(10\ s)$, at a location depending on the bubble size and substrate temperature. Beyond this critical timescale, bubbles exhibited a state of partially frozen equilibria, where the top portion of the bubble remained liquid (see the second row in Fig. 4b). The one exception to this trend of partial freezing was for $\Omega = 5\ \mu L$ and

$T_w = -40\ °C$, in which case bubbles were able to completely freeze (see the first row in Fig. 4b). In contrast to the bottom-up freezing of droplets[22], completely frozen bubbles did not exhibit a pointy tip, as instead the water can expand within the hollow interior.

The observed trend of incomplete freeze fronts can be rationalized by the poor thermal conductance of the long and slender soap film. Conservation of energy within the bubble's film can be expressed as $dT/dt = \alpha \nabla^2 T$, where $\alpha$ is the thermal diffusivity corresponding to water or ice. The thermal diffusion timescale for a bubble of radius $R$ scales as $\tau_D \sim R^2/\alpha_i$. In contrast, the timescale of the freeze front motion[22], $\tau_f \sim R/v$, scales as $\tau_f \sim \tau_D L/(c_i \Delta T_i)$, where $c_i = 2.027 \times 10^3\ J\ kg^{-1}\ K^{-1}$ is the specific heat capacity of ice and $\Delta T_i \sim 10\ K$ is a typical temperature difference. The diffusion timescale is about an order of magnitude smaller than the freezing timescale, $\tau_f \sim 10\ \tau_D$, which allows us to assume a quasi-steady temperature profile: $\nabla^2 T = 0$. This remains true for water, as its heat capacity is about two times higher than that of ice.

Given the thin film thickness of the bubbles, it is assumed that a deposited bubble is predominantly cooled by the surrounding air rather than by conduction into the substrate. The temperature field can therefore be obtained by numerically modeling the natural convection occurring in the air above the chilled substrate (see Supplementary Note 1). A thermal boundary layer thickness of $\zeta \approx 3$ cm was obtained, in agreement with experimental measurements, using a thermocouple and translation stage (Supplementary Fig. 7). The air temperature profile was slightly modified by the presence of a bubble, which was also captured in the simulation (Fig. 4f and Supplementary Figs. 8 and 9).

The velocity of freeze fronts is limited by the latent heat of fusion being released. Our model assumes that this latent heat, $\dot{Q}_{LH}$, is mostly dumped into the substrate via conduction across the frozen portion of the bubble, $\dot{Q}_i$, as both the thermal conductivity and temperature gradient of the ice are much larger than the surrounding air. The liquid upper portion of the bubble is also conducting heat, $\dot{Q}_l$, either toward or away from the freeze front, depending on the direction of the temperature gradient. We neglect convection in the liquid portion of the bubble, as the Marangoni flow has typically dissipated by the time the freeze front has grown appreciably. This balance of heat in vs. heat out at the freeze front can be summarized as $\dot{Q}_{LH} + \dot{Q}_l = \dot{Q}_i$, where $\dot{Q}_l$ can be negative in some cases. For a bubble of fixed radius of curvature $R$, these terms can be fully expressed as

$$\left[ \rho L R \frac{d\psi}{dt} + k_l \frac{\Delta T_l}{R(\pi - \psi)} \right] (2\pi Re_0 \sin\psi) \sim k_i \frac{\Delta T_i}{R(\psi - \psi_0)} (2\pi Re_0 \sin\psi), \quad (6)$$

where $k_i$ and $k_l$ are the thermal conductivities of ice and liquid, $\Delta T_i$ and $\Delta T_l$ are the temperature differences across the frozen and unfrozen portions of the bubble, $\psi_0$ is the fixed angular coordinate of the bubble's contact line (see Fig. 4c), and $\psi$ is the angular coordinate of the evolving freeze front. The slight difference in density between water and ice was neglected here, such that $\rho = \rho_l \approx \rho_i$. The freeze front is always at the melting temperature, $T_m$, while the contact line is always at the substrate temperature, $T_w$, resulting in a temperature difference of $\Delta T_i = T_m - T_w$ across the frozen portion of the bubble. The temperature at the top of the bubble, $T_t$, was found from the numerical simulations of the temperature field, such that $\Delta T_l = T_t - T_m$ across the liquid portion of the bubble. For the 5-μL bubbles, the liquid film tended to conduct heat away from the freeze front ($T_t < T_m$), while heat was conducted into the freeze front for the 500-μL bubbles ($T_t > T_m$, see Supplementary Fig. 9a).

Canceling like terms and using a dimensionless time $t^* = t/\tau_f$, Eq. 6 can be non-dimensionalized:

$$\frac{d\psi}{dt^*} \approx \frac{\beta_1}{\psi - \psi_0} - \frac{k_l}{k_i} \frac{\Delta T_l}{\Delta T_i} \left( \frac{\beta_2}{\pi - \psi} \right), \quad (7)$$

where $\beta_1$ and $\beta_2$ are geometrical pre-factors. As seen in Fig. 4d and Supplementary Fig. 10a, the arc length of the growing freeze front, $S(t) = R\psi(t)$, is captured by Eq. 7 for fixed values of $\beta_1 = 1.8$ and $\beta_2 = 30$. Therefore, all of the data collapse onto a universal nondimensional curve, as shown in Fig. 4e.

The freeze front stops propagating as $d\psi/dt^* \rightarrow 0$ at a critical angular coordinate $\psi \rightarrow \psi_f$ ($S \rightarrow S_f$). The energy equation is then simplified to $\dot{Q}_l = \dot{Q}_i$, such that $\psi_f$ can be found as

$$\psi_f = \pi - (\pi - \psi_0) \left( \frac{\beta_1 k_i}{\beta_2 k_l} \left( \frac{\Delta T_i}{\Delta T_l} \right) + 1 \right)^{-1}. \quad (8)$$

This can also be expressed as a critical height, $h_f = 2R\sin((\psi_f - \psi_0)/2)$ (see Supplementary Fig. 10b). Equation 8 is plotted in Fig. 5, where the slope corresponds to $\beta_1 k_i/\beta_2 k_l = 0.35$. This

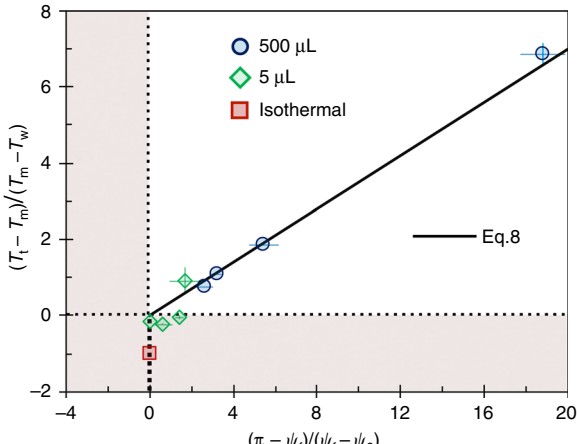

**Fig. 5** Partial freezing of bubbles. The critical angle at which the freeze front completely stops was found by balancing the conduction of heat across the icy and liquid portions of the bubble (Eq. 8). The isothermal condition corresponds to the vertical dashed line, where complete freezing occurs ($\psi_f = \pi$, square data point)

results in $k_i/k_l = 5.83$ (for the same $\beta_1$ and $\beta_2$ as before), which is close to $k_i/k_l = 3.93$ corresponding to that of pure water. This discrepancy is due to the existence of glycerol, soap solution, and low temperatures. Three distinct regimes of freezing are possible: the nonfrozen regime, where $T_w \geq T_m$ and $\psi_f \rightarrow \psi_0$, the completely frozen regime, where $T_t \leq T_m$ such that $\psi_f \rightarrow \pi$, and the partially frozen regime, where $T_t > T_m > T_w$ and $\psi_0 < \psi_f < \pi$ (where $\psi_f$ is found by Eq. 8). Across all bubble volumes and surface temperatures, the experimental values of $\psi_f$ collapse perfectly onto this curve, validating the model. The experimental measurements of $\psi_f$ were complicated by the continued growth of frost on the substrate at a velocity of order 1 μm s⁻¹ (Fig. 4g and Supplementary Figs. 11 and 12). This rate of frost growth is about two orders of magnitude slower than that of the freezing front, and can therefore be neglected aside from noting that the location of $h_f$ slowly translates upward with the growing frost.

Marginal regeneration: Once a bubble reached its partially frozen equilibrium, initially there was no appreciable flow in the upper liquid portion of the bubble. After about 100 s, there was a sudden reappearance of plumes within the liquid dome (see Supplementary Fig. 5b). In contrast to the Marangoni freezing-induced plumes that were observed on initial deposition, these new plumes were because of marginal regeneration. Specifically, the ice-liquid boundary continually thickened at the expense of the top of the liquid dome due to drainage. This was visually evident from the appearance of interference fringes on the thinning liquid dome. The timescale of the formation of these plumes is consistent with the drainage timescale: $t_d \sim (\eta R)/(\rho g b^2)$ $\sim 10^2 - 10^3$ s for $R \sim 1 - 10$ mm[8].

Collapse: After $\mathcal{O}(10$ min) of partially frozen equilibrium, the liquid dome suddenly deflated and collapsed (Fig. 6a, b). The timescale from beginning to end of the collapse ranged from ~0.1 to 10 s, depending on the trial. This gradual deflation of the liquid dome over a span of several seconds is in sharp contrast to the dynamics of drainage-induced failure, where a hole opens and rapidly spreads (Supplementary Fig. 13). One possibility for the collapse is that the gradual cooling of the air within the bubble leads to a reduced internal pressure, as per Gay-Lussac's law[8]. However, for the typical value of thermal diffusivity of water/vapor with respect to air, $\alpha_v \sim 10^{-5}$ m² s⁻¹, the thermal diffusion timescale of $\tau_D \sim R^2/\alpha_v \sim 0.1$ s is too fast, given that the liquid remains dome-shaped for about 10 min.

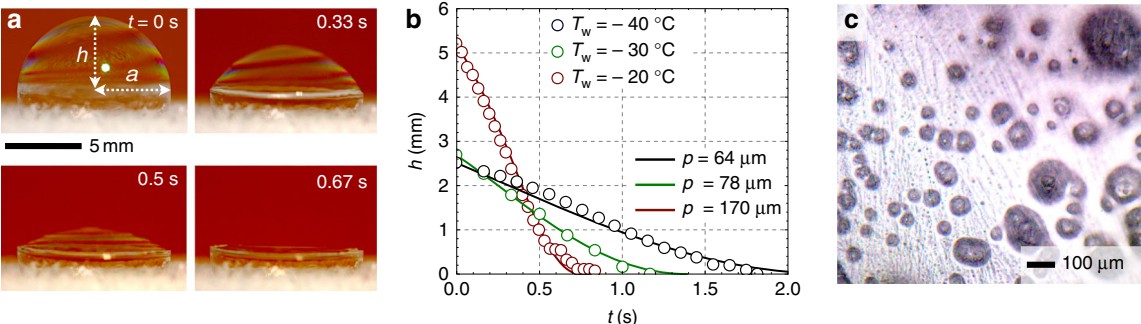

**Fig. 6** Collapse of the liquid dome of partially frozen bubbles. **a** The sudden collapse of the liquid roof of a partially frozen $\Omega = 500$-μL bubble. Time zero corresponds to the beginning of dome collapse, which was completed in ~1 s. Conditions were $T_w = -20\,°C$, $T_\infty = 24.6\,°C$, and RH = 58.8%. **b** The height of the liquid dome ($h$) against time for different substrate temperatures, for $\Omega = 500$-μL bubbles. Solid lines correspond to the theoretical drainage model provided by Eq. 10, for the best-fit values of $p$ shown in the legend. **c** Microscopy revealed discontinuities on the frozen portion of the bubble (~10–100 μm), responsible for gradually draining out the pressurized air from within the bubble

Instead, consider the positive Laplace pressure of the air within the bubble, due to the convex curvature of the liquid dome. This pressure difference is given by $\Delta P = 4\gamma/r$, where $r = (a^2 + h^2)/2h$ is the liquid's radius of curvature, $a$ is the fixed contact radius of the liquid/ice interface, and $h$ is the height of the liquid dome (see Fig. 6a). If the frozen portion of the bubble included small pores, this Laplace pressure would cause air to flow out of the pores with a dynamic pressure of $\frac{1}{2}\rho_{air}v_{air}^2$. Following Bernoulli's law, we can equate the dynamic pressure with the Laplace pressure to obtain

$$v_{air} \approx \sqrt{\frac{16\gamma h}{\rho_{air}(a^2 + h^2)}}. \tag{9}$$

By mass conservation, $-dV_s/dt \sim \pi p^2 v_{air}$, where $V_s = (\pi h/6)(3a^2 + h^2)$ is the volume of the liquid spherical cap and $p$ is the pore radius. Taking the derivative of the volume with respect to $h$, the change in height of the liquid dome with respect to time is

$$dh/dt \approx -8p^2 \sqrt{\frac{\gamma h}{\rho_{air}(a^2 + h^2)^3}}. \tag{10}$$

The density of the air inside a bubble was found by using the ideal gas law, $\rho_{air} = P/(R_s T_{air})$, where $P$ is the absolute pressure, $R_s = 287.058\,\mathrm{J\,kg^{-1}\,K^{-1}}$ is the specific gas constant, and $T_{air}$ is the average temperature inside the bubble, which was calculated from the computational results (Supplementary Fig. 9). These values of $T_{air}$ were used to obtain $\rho_{air} \approx 1.27\,\mathrm{kg\,m^{-3}}$, $1.28\,\mathrm{kg\,m^{-3}}$, and $1.29\,\mathrm{kg\,m^{-3}}$ for different substrate temperatures of $T_w = -20\,°C$, $-30\,°C$, and $-40\,°C$, respectively. Choosing values for $p$ that obtained a best fit to the experimental data results in $p \sim 10–100\,\mu m$ (Fig. 6b), consistent with the size of porous features observed within the ice (see Fig. 6c).

As evidenced by Fig. 6c, it is possible that multiple pores exist, in which case $p^2 = Np_{avg}^2$ where $N$ is the number of pores and $p_{avg}$ is the average pore diameter. For a minority of the room-temperature experiments, the liquid dome of the partially frozen bubble ruptured before the Laplace-induced collapse could occur (see Supplementary Fig. 13). The timescale of bubble rupture (~1 ms) is much faster than the collapse event discussed here, further demonstrating that the collapse mechanism is fundamentally different from film rupture.

Figure 7 summarizes every mode of freezing in a comprehensive phase map for any possible ambient condition. The unfrozen region corresponds to the case where the substrate temperature is warmer than the melting temperature of the soap solution, preventing the heterogeneous nucleation of a bottom-up freeze front. Partially frozen bubbles occur when the top of the bubble

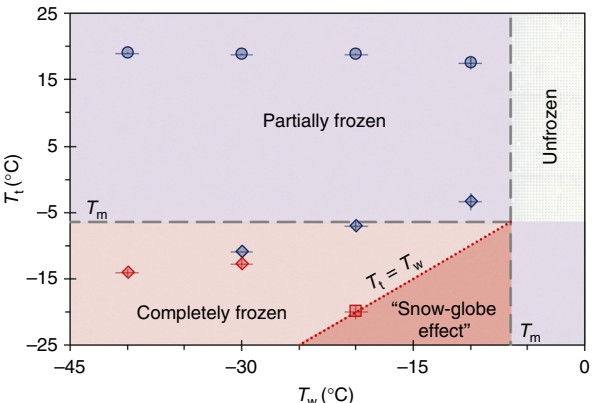

**Fig. 7** Regime map of different kinds of freezing behavior in soap bubbles. Bubbles completely freeze in the red regions ($T_t < T_m$ and $T_w < T_m$), while a partially frozen equilibrium occurs for bubbles in the purple area ($T_t > T_m$). The dark red region is a subset of the "Completely Frozen" regime, where the Marangoni freezing is able to produce the "Snow-Globe Effect" due to the added criterion of $T_t \leq T_w$. Blue and red data points correspond to experimentally observed partially frozen or completely frozen bubbles, respectively, where the enclosed air volumes were 5 μL (diamonds), 500 μL (circles), or 10 mL (squares). Long dashed lines correspond to $T_t = T_m$ and $T_w = T_m$, while the dotted line corresponds to using an isothermal freezer

is warmer than the melting temperature, while the substrate temperature is below. Completely frozen bubbles are observed when both the surface temperature and the temperature at the top of the bubble are subfreezing ($T_t < T_m$ and $T_w < T_m$). In this region, the "Snow-Globe Effect" is observed when $dT/dz \leq 0$, or in other words, $T_t \leq T_w$ (see the diagonal line and the dark red region). This condition is satisfied inside the walk-in freezer, where the substrate temperature was equal to the temperature at the top of the bubble. Note that all types of freezing tended to produce some degree of flow due to Marangoni freezing, even though the "Snow-Globe Effect" only occurred for the isothermal case.

Finally, control tests were performed with a 1% concentration of a pure surfactant, sodium dodecyl sulfate (SDS), rather than 1% dish soap. Bubbles with SDS bursted within $\mathcal{O}(10\,s)$ upon deposition on an icy substrate in room-temperature conditions (Supplementary Fig. 14), most likely due to drainage. This drainage timescale is consistent with a previous report[8]. During the short time where the bubble was intact, the bottom-up freezing dynamics were equivalent to that with the dish soap. For the isothermal conditions, $\Omega = 500$-μL bubbles containing SDS burst within $\mathcal{O}(1\,s)$.

In conclusion, the freezing dynamics of soap bubbles are multifaceted and fundamentally distinct from the classically studied scenario of freezing bulk volumes of liquid. Under chilled and isothermal conditions, hundreds of ice particles are detached from the freeze front and swirl around the bubble in a "Snow-Globe Effect." This beautiful dance was caused by the local input of latent heat at the freeze front, causing a strong Marangoni flow capable of breaking off small ice crystals. Isothermal bubbles therefore freeze very efficiently due to hundreds of fronts growing in tandem and interlocking together. When a bubble is frozen on a cold stage in a room-temperature environment, the freeze front slowly propagates upward and comes to a complete stop at a critical height. The freeze-front dynamics were found to be captured by a Stefan problem governed by a balance between latent heat and conduction across the frozen and unfrozen portions of the bubble. After the halting of the freeze front, the partially frozen bubble remains in equilibrium for many minutes, followed by the deflation and collapse of the liquid dome, due to its Laplace pressure forcing air through small pores in the ice. These findings show that the dynamics of freezing liquid is highly dependent on its geometric conditions, and that a rich variety of multiphase phenomena occur when a liquid volume is neither continuous nor surface-bound.

## Methods

**Materials**. The soap bubbles were generated using a solution consisting of 79% (by volume) distilled water, 20% glycerol (Sigma-Aldrich, 56-81-5), and ≈1% dish soap (Palmolive®, Ultra AntiBacterial Dish Liquid). The dish soap consists of 98% inert ingredients (water, sodium laureth sulfate, lauramidopropyl betaine, sodium dodecylbenzene sulfonate, SD 3 A alcohol, sodium xylene sulfonate, fragrance, tetrasodium EDTA, and dyes), and 2% active ingredient (L-lactic acid). For the control experiments, instead of dish soap, sodium dodecyl sulfate (Sigma-Aldrich, 75746) was used. After stirring the mixture together, it was allowed to sit overnight before running any experiments. Neglecting the effects of the soap, the freezing temperature of an 80–20% water/glycerol mixture was previously reported[29] to be $T_m \approx -6.5\,°C$.

For the room-temperature experiments, bubbles were deposited on a frosted Peltier stage (ramé-hart, Model 100-30) using a syringe pump (ramé-hart, Model 100-22). A needle with inner and outer diameters of 0.7 mm and 2.1 mm, respectively, was used. Experiments were recorded using a DSLR camera (Canon®, EOS 5D Mark III) with a macro lens (Canon MP-E 65 mm f/2.8 1–5×). After freezing, the pores and/or bubbles trapped within the ice were characterized using a top-down optical microscope (Nikon 150LV) with a long working-distance lens (Mitutuyo, MPlan APO).

For the isothermal experiments, a polystyrene Petri dish (VWR, 25384-326) was filled with distilled water and left in a freezer (Frigidaire, Model FRT21IL6JB2) with a temperature of −20 °C overnight. Ice disks were kept within a cooler (Igloo®) when transporting to the walk-in freezer (Conviron, Model C1008). An ice disk was allowed to sit within the walk-in freezer for at least an hour before beginning experiments, to ensure that its temperature was that of the air. For some control experiments, the bubbles were deposited on a dry silicon substrate rather than an ice disk. Experiments were recorded using a high-speed camera (Vision Research, Phantom v711). The relative humidity and air temperature of the walk-in freezer were measured by a hygrometer (E + E Elektronik, Model EE210). Thermal imaging experiments were conducted using an IR camera (FLIR SC655).

**Lighting**. For the room-temperature experiments, a spotlight was used for front lighting (Advanced Illumination, Model SL164), while a square LED was used for a backlight (Advanced Illumination, Model BX0808). When imaging a frozen bubble with the top-down microscope, LED lighting was used to minimize heating effects (Nikon, LV-UEPI Universal Epi Illuminator 2). For the isothermal experiments, a round LED light (Genaray, Model SP-AD75) was placed underneath the ice disk with a horizontal orientation to illuminate the bubble (see Supplementary Fig. 3a). Plumes (Fig. 3 and Supplementary Fig. 5a) were visualized using a LOWEL DP light which was kept about 5 m away from the experimental setup to minimize heating effects.

**Image processing**. Videos were imported to an open-source software (Tracker) to track three points (leftmost, rightmost, and center) of the freezing bubbles. The coordinate system was placed such that the $y$ axis crossed the center of the bubble, while the $x$ axis was placed at the contact line of the bubble/substrate interface. The final angle at which the freeze front stopped propagating, was found by importing the corresponding image to ImageJ.

## Data availability

The data supporting the findings of this study are available within the article and the associated Supplementary Materials. Any other data are available from the corresponding author upon request.

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

## Acknowledgements
We thank Yanqing Fu for introducing us to the online videos of freezing bubbles. We acknowledge R. Mills and S. Case for the use of their IR camera and D. Mitchem for granting access to the walk-in freezer. Special thanks to J. Lloyd, B. Chang, R. Mukherjee, and H. Park for their help with some of the freezing bubble photography. S. N. acknowledges the support of the European Union's Horizon 2020 research and innovation program LubISS (Marie Sklodowska-Curie grant agreement No 722497). This work was supported by startup funds from the Department of Biomedical Engineering and Mechanics at Virginia Tech.

## Author contributions
S.F.A., S.N., and J.B.B. designed the research. S.F.A. and C.M.K. carried out the experiments. S.F.A. analyzed the data. S.F.A., S.N., P.Y., and J.B.B. developed the theoretical modeling. S.F.A. and P.Y. performed the numerical simulation. S.F.A., S.N., and J.B.B. wrote the paper. All authors proofread, made comments, and approved the paper.

## Additional information

**Competing interests:** The authors declare no competing interests.

