## [Peer Review File · Nature Communications]

Reviewer #1 (Remarks to the Author):

Comments on "How bubbles freeze" by Ahmadi et al.

The authors present a detailed and interesting study on the physics of freezing soap bubbles placed on a chilled icy substrate. Two scenarios were identified: 1) for an isothermally supercooled bubbles, icy crystals are entrained by Marangoni flow as nucleation point on the bubble cap for freezing; 2) when the ambient temperature is higher than the melting point, the freeze front eventually halts at a critical height, followed by the deflation and collapse of the liquid dome. The authors also characterized the dynamics of the Marangoni flows and the freeze fronts in the related scenarios. Overall, I believe the article will be of interest to the broad audience, but I have the following concerns that require to be appropriately addressed before I can recommend publication.

1. page 3, what is the meaning of Ω ? The liquid volume forming the bubble? It should be specified.
2. The authors used a commercial dish soap with unknown components. Any particular reason not using a common surfactant? Any effects of the dish soap on the freezing temperature of the water/glycerol solution?
3. Following the last question, how the dish soap effects the Marangoni flow? Does the Marangoni flow only introduce by temperature gradient?
4. Eq. (1), the Taylor-Culick law only fits with the free slip boundary condition. Can the authors comment on/justify that it is still applicable with a surfactant-laden interface here? 1% dish soap suggests a quite high surfactant concentration already.
5. Page 5, no caption of Fig. 2b is given. In Fig. 2b, the third row, why are those ripples formed on the bubble surface with $\Omega=5 \frac{\mu\text{L}}{\text{L}}$, but not on other larger bubbles?
6. page 7, when the authors estimated the total Marangoni force, should the effect of the dish soap be considered?
7. page 9, "two phase" should be "two-phase"; "is the is" should be "is the".
8. page 10, for the estimated tip radius, the authors got the value from the initial film thickness or the image analysis? Did the two approaches give a consistent value?
9. Eq. (6), does it assume the air inside the bubble also has the same temperature as the ambient? Can the authors comment on this assumption?
10. What does the symbol [t!] mean at the right corner of Fig. 4g?
11. page 12, for the simulation (fig. S5), how did the authors model the bubble? Is there any convection effect for the air inside the bubble?
12. Fig. 6, is there a case where the liquid film ruptures instead of collapse of the liquid dome?
13. Eq. (11), the numeric factor should be 16 instead of 8?
14. In the SI, where is Fig. S11? I could not find it anywhere although it was mentioned in the caption of Fig. S9.

Reviewer #2 (Remarks to the Author):

The authors studied several distinct different icing regimes of the soap bubble by combining scaling analysis, experiments and numerical methods.

Two different situations were investigated: For isothermally super-cooled bubbles, ice particles are detached from the freeze front and swirl around the bubble; For the situation when the ambient is warmer than the melting temperature, the freeze front slowly propagates upward and comes to a complete stop at a critical height.

In the end of the paper, they used a regime map to summarize the different kinds of freezing behaviors in soap bubbles.

The idea is novel and original, and the results are impressive. The paper was well prepared. I do have some concerns on the manuscript. I would like to see an improved version, in which the issues/comments below have to be satisfactorily addressed.

Major comments:

1. Figure 2 and the corresponding discussions: This is a trivial result for film thickness determination. I do not know whether it should be in the main part of the paper. The authors may consider to move it to the supplementary material or the method section.

Though it is not a novel result, the details of the calculations are still missing.

For the Fig 2a, it only records the early stage of the bursting of bubbles. If the authors continue to record the evolution of the L , the scaling will be changed (like the green line). Will the film thickness change with time?

For the Fig 2b, how to calculate the value of L of the bubble when there is a splash at the edge?

2. In the Fig 3a, the contact angle of the bubble is changed after being deposited on an ice disk. How to keep the contact angle and the shape of bubbles to be similar on the icy substrate? Will the crystal structure of ice influence the roughness of the substrate and the shape of the bubble?

What causes the different areas in the bubble base in the first and second images in Fig 3a?

3. Fig 3g: as the tip is pointed, how could the authors define the tip radius of the ice layer?

4. Page 7: how to obtain the physical properties of the soap solution, such as the latent heat?

5. Page 7, 'dissipates by thermal conduction across the liquid film...'

What about the convective motion? The authors have to elaborate more on this point.

6. Page 8, the last paragraph 'Marangoni plumes must be detaching ice ... crack invitation ...'

It does not read clear to me. The authors have to elaborate more on this part.

7. Page 9, 'Comparing the pull-off force and inertia ($F_i > F_{\text{crack}}$) reveals that a dendrite must be smaller than $l < 10\text{nm}$ for detachment. This is consistent with the observation that detachment events were too small to directly visualize; rather, they were deduced from the flowing crystals later growing to a visible (i.e. micro-scale) size.'

This part is too farfetched! I am not convinced by the estimation of 10 nm, and the authors cannot conclude this number to be corrected because you cannot visualize it. More discussions and revisions have to be provided here. The authors have to be more rigorous.

8. Page 10, 'This growth rate of $v_i \sim 100 \mu\text{m/s}$ is in agreement with the predicted detachment size of $l < 10\text{nm}$ for dendrites,...'

Again, this is too farfetched! What is the direct connection between v_i and the detachment size? This argument is really unconvincing.

9. The heat transfer efficiency between air and liquid is different from that of air and ice. In Fig 4f, have the authors considered this effect?

Minor comments:

1. Fig 2 caption: 'curves seen in a' -> 'curves seen in b'.
2. Fig 4g: what's the meaning of the [t !] in the bottom right corner ?
3. Fig 6 caption: ' ^oC' should be added after 'T_infinite = 25.56'.
4. Fig 6c: Eq. 5 -> Eq. 12.
5. Page 7, 13.5 ^oC - > 13.5 K.

Reviewer #3 (Remarks to the Author):

Comments:

This work describes experimental and theoretic results on the interesting phenomenon of bubble freezing.

Specifically, the authors describe two discrete physical processes that govern the freezing of a bubble in iso-thermal and non-iso-thermal ambient conditions.

The experimental data produced in the manuscript is of high quality and the results are explained clearly.

In addition, the scaling analysis and numerical computations augment the ideas derived from the experiments.

Overall, the paper is well-written and I think it would of general interest to the readers of the journal. I would recommend the work for publication.

However, there are a few comments and questions which authors must address in the revised manuscript.

A) The abstract begins with a sentence ``\emph{The two-stage freezing process of liquid droplets and films is well known}...'. I am certain that a large number of readers will not be an expert in this field, so a reference to the two-stage freezing process is needed.

B) page 5: ``\emph{These thermal plumes exhibited a characteristic radius of $R_p \sim 1$ mm}...'. As I understand, a plume is an ascending/descending column of a fluid into another. In that case will

it not be sensible to choose the height of the plumes visible in the experimental images as the characteristic length scale rather than R_p ?

C) Caption of figure 3: ``\emph{...due to latent heat.} \rightarrow ...due to the release of latent heat.

D) page 7: The authors make the statement that ``\emph{... such that Marangoni flow can be induced entirely within a plume}', is this accurate? Do you expect a strong thermal gradient of 13°C within the plume? A closer look at the thermographic images does not support the statement.

E) page 7: The authors use a thermal diffusive length scale δ in the calculation of the viscous drag F_η , what is the justification for this particular choice. Instead, the thickness of the bubble e_0 appears to be more appropriate/natural scale.

F) page 8: Choice of words ``\emph{... the inertia of the Marangoni flow (F_i) ...}'. This terminology is confusing, what do authors want to refer here, inertial or Marangoni forces?

**Response to Referees for NCOMMS-18-33887-T:
“How bubbles freeze”**

Reviewer #1:

The authors present a detailed and interesting study on the physics of freezing soap bubbles placed on a chilled icy substrate. Two scenarios were identified: 1) for an isothermally supercooled bubbles, icy crystals are entrained by Marangoni flow as nucleation point on the bubble cap for freezing; 2) when the ambient temperature is higher than the melting point, the freeze front eventually halts at a critical height, followed by the deflation and collapse of the liquid dome. The authors also characterized the dynamics of the Marangoni flows and the freeze fronts in the related scenarios. Overall, I believe the article will be of interest to the broad audience, but I have the following concerns that require to be appropriately addressed before I can recommend publication.

We thank the reviewer for the encouraging feedback and have responded to all comments below. All changes to the revised manuscript corresponding to Reviewer #1’s feedback have been highlighted in yellow.

1. page 3, what is the meaning of Ω ? The liquid volume forming the bubble? It should be specified.

We thank the reviewer for pointing this out. We have now clarified that the Ω is the volume of the air enclosed by the liquid film.

Changes to page 3

an icy substrate, such that the freezing process could begin immediately (Fig.S1). For isothermal experiments, soap bubbles enclosing air volumes of $\Omega = 500 \mu\text{L}$ or 10 mL were

2. The authors used a commercial dish soap with unknown components. Any particular reason not using a common surfactant? Any effects of the dish soap on the freezing temperature of the water/glycerol solution?

The reasons behind using a commercial dish soap were as follows:

- a) There is good precedent for using commercial dish soap as the surfactant for research on soap films/bubbles. For instance, see the following papers:
 - i. Cohen, C., Texier, B.D., Reyssat, E., Snoeijer, J.H., Quéré, D. and Clanet, C., 2017. On the shape of giant soap bubbles. *PNAS*, 114, (2017): 2515-2519.
 - ii. Lhuissier, H. and Villermaux, E. “Soap films burst like flapping flags.” *Phys. Rev. Lett.* 103, (2009): 054501.
 - iii. Isenberg, C. “Soap films and bubbles.” *Phys. Educ.* 16, (1981): 218.
 - iv. Skotheim, J.M. and Bush, J.W. “Evaporatively Driven Convection in a Draining Soap Film.” *Phys. Fluids* 12, (2000): S3-S3.
 - v. Shen, L., Denner, F., Morgan, N., van Wachem, B. and Dini, D. “Before the bubble ruptures.” *Phys. Rev. Fluids* 2, (2017): 090505.

- vi. Rosi, T., Gratton, L.M., Onorato, P. and Oss, S. “Light interference from a soap film: a revisited quasi-monochromatic experiment.” *Phys. Educ.* 54, (2018): 015018.

On a related note, we have decided to revise the title of the revised manuscript, changing ‘bubbles’ to ‘soap bubbles,’ to ensure that nobody mistakes our bubbles for boiling rather than surfactant-based.

Changes to the title

How soap bubbles freeze

- b) We have added more information to the paper clarifying our soap solution:

Changes to page 23

Ultra AntiBacterial Dish Liquid). The dish soap consists of 98% inert ingredients (water,

Changes to page 24

sodium laureth sulfate, lauramidopropyl betaine, sodium dodecylbenzene sulfonate, SD 3A alcohol, sodium xylene sulfonate, fragrance, tetrasodium EDTA, and dyes), and 2% active ingredient (L-Lactic acid). For the control experiments, instead of dish soap, sodium dodecyl sulfate (Sigma-Aldrich, 75746) was used. After stirring the mixture together, it was allowed

- c) In the revised manuscript, we have performed new experiments using a pure surfactant instead of dish soap. Specifically, our new solution consists of 79% water, 20% glycerol, and 1% Sodium Dodecyl Sulfate (SDS). We found that bubbles burst within $\mathcal{O}(10\text{ s})$ after depositing on a chilled substrate in a room temperature environment. The bottom-up freezing process was identical to that using dish soap, but now the bursting occurred so quickly that we couldn’t analyze the partially frozen equilibria or bubble deflation. This is now clarified in the main manuscript and in a new supporting figure:

Changes to page 22

Finally, control tests were performed with a 1% concentration of a pure surfactant, sodium dodecyl sulfate (SDS), rather than 1% dish soap. Bubbles with SDS bursted within $\mathcal{O}(10\text{ s})$ upon deposition on an icy substrate in room temperature conditions (Fig. S14), most likely due to drainage. This drainage time scale is consistent with a previous report⁸. During the short time where the bubble was intact, the bottom-up freezing dynamics were equivalent to that with the dish soap. For the isothermal conditions, $\Omega = 500\ \mu\text{L}$ bubbles containing SDS burst within $\mathcal{O}(1\text{ s})$.

Changes to Figure S14

12 Control Experiments Using a Pure Surfactant

Figure S14: (A) Freezing of a $\Omega = 500\ \mu\text{L}$ bubble that was made using a solution containing 1% SDS, rather than dish soap, as the surfactant. The bubble was deposited on an icy substrate with a temperature of $T_w = -20 \pm 1\ ^\circ\text{C}$ in an environment of $T_\infty = 22.1\ ^\circ\text{C}$ and $RH = 13.7\%$. Arrows show the location of the freeze front. Bubbles with 1% SDS mostly burst within $\mathcal{O}(10\text{ s})$. Drainage induced thinning of the bubble from top was responsible for the bursting of the bubble. (B) High-speed imaging showing the rapid bursting of a bubble at the end of its lifetime.

This is in a good agreement with Champougny et al. (Ref. 8 in revised manuscript) that showed an average lifetime of $\mathcal{O}(10\text{ s})$ for 0.8 mL bubbles with a pure surfactant. So, in short, the freezing processes are indeed similar for a pure surfactant but the bubble lifetimes are not adequate for full characterization. We feel this justifies our choice of dish soap.

- d) Our solution consists of just 1% dish soap which we believe does not appreciably affect the freezing temperature of the water/glycerol mixture.

3. Following the last question, how the dish soap effects the Marangoni flow? Does the Marangoni flow only introduce by temperature gradient?

We expect that the Marangoni flow is being primarily driven by the temperature gradient and not by the surfactant. A justification for this assumption is provided in the revised manuscript:

Changes to page 6-7

Possible Mechanisms. The mechanism for the vertical plumes that emerge when the bubble contacts the icy surface is far from obvious. At least four types of flows can be envisaged: Marangoni flow due to non-uniform surfactant concentrations, flow due to non-uniform surface curvature (marginal regeneration), buoyant flow due to thermal effects, or a thermal Marangoni flow induced by the latent heat of fusion. The first two flow mechanisms do not require a temperature gradient and can therefore be evaluated using bubbles deposited on a dry substrate at room temperature conditions (Fig. 3a). The distribution of surfactant along the interface can be isolated by considering a bulk pendant drop suspended in air, where about 40 min was required to achieve a steady-state surface tension (Fig. S1). However, recall that our soap bubbles are only about $1\ \mu\text{m}$ in thickness. For such thin films, having a large surface-area-to-volume ratio, the diffusion of surfactant to the free surface should be much faster³³. For example, the ratio of diffusive time scales between the pendant drop

and bubble is $R_{\text{drop}}^2/\epsilon_0^2 \sim 10^6$, such that a steady-state surface tension should be achieved within about 1 ms for the bubble. In contrast, we observed that room-temperature bubbles on dry substrates generated plumes over a very long time scale of $\mathcal{O}(10\ \text{min})$ (Fig. S5), ruling out asymmetric surfactant concentrations as a likely mechanism. Besides, the transient effects in surface tension measured with the pendant drop were temporal in nature, with no appreciable spatial gradient required to generate a flow along the interface.

4. Eq. (1), the Taylor-Culick law only fits with the free slip boundary condition. Can the authors comment on/justify that it is still applicable with a surfactant-laden interface here? 1% dish soap suggests a quite high surfactant concentration already.

The Taylor-Culick law has been already used in the literature even for bubbles with concentrations of dish soap up to 40% (Ref. 32 in revised manuscript). Given our concentration of dish soap is only 1%, we therefore feel that it is valid to use the Taylor-Culick law.

5. Page 5, no caption of Fig. 2b is given. In Fig. 2b, the third row, why are those ripples formed on the bubble surface with $\Omega = 5 \mu\text{L}$, but not on other larger bubbles?

There was a caption for Fig. 2b but it was accidentally labeled as ‘a’ again. We thank the reviewer for catching this typo and have fixed it. Please note that we have moved this original Fig. 2 to the SI as the second reviewer suggested.

Regarding the ripples, these are not solely associated with the small bubble. These flows are due to an effect known as marginal regeneration, driven by a mismatch in curvature between the wetting meniscus and the arc of the bubble. With proper lighting, flow due to marginal regeneration is also visible in the larger bubbles. We have added a discussion on marginal regeneration in the revised manuscript. We also add new analysis that shows that, in the case of freezing, the flow is due to Marangoni freezing and not marginal regeneration. Moreover, we have added two figures (Figs. 3a and S5) in the revised manuscript that clarify these flows for the control case of all-room-temperature conditions. Please see our response to Review #3 below, who had several questions about the flows observed in the bubbles, where we paste these new figures and text in full.

6. page 7, when the authors estimated the total Marangoni force, should the effect of the dish soap be considered?

See our response to comment (3) above.

7. page 9, “two phase” should be “two-phase”; “is the is” should be “is the”.

The “two phase” phrasing as been fixed as shown below. The “is the” phrase no longer exists as the revision has removed this sentence.

Changes to page 12

The growth rate of the ice front can be modeled by using the well-known two-phase Stefan problem where a semi-infinite ($0 < y < \infty$) supercooled liquid with temperature $T_l < T_m$ is exposed to a temperature T_c at its boundary ($y=0$) at time zero. The tip velocity is then
--

8. page 10, for the estimated tip radius, the authors got the value from the initial film thickness or the image analysis? Did the two approaches give a consistent value?

We obtained it from image analysis, doing a best-fit to the tip radius shown in Fig. 3g. This measurement was in agreement with the film thickness to within a factor of about 2. This value is also consistent with the earlier report (see Table 4 in reference 28 of the revised manuscript) for the tip radius of ice layer. We have clarified these points in the revised manuscript:

Changes to page 12

where ν is the ratio of the thermal diffusivity of ice to the water defined as $\nu = \sqrt{\alpha_i/\alpha_l}$. The tip diameter was crudely measured as $2R_i \approx 1.28 \mu\text{m}$ (Fig. 2f), consistent with a previous report²⁸ and the need to be contained within the film ($e_0 \approx 1.7 \mu\text{m}$, see Fig. S2a). Plugging this value of R_i into Eq. 5 gives a theoretical tip velocity of $v_i \approx 250 \mu\text{m/s}$, in agreement with experimental growth measurements of $v \approx 300 \mu\text{m/s}$ for the bottom-up freeze front and $v_i \approx 250 \mu\text{m/s}$ for the crystals suspended within the bubble film (Fig. 2f).

9. Eq. (6), does it assume the air inside the bubble also has the same temperature as the ambient? Can the authors comment on this assumption?

Our original manuscript assumed room temperature air inside the bubble. We agree in hindsight that this is a poor assumption, and the revised manuscript now averages the temperature profile from within the bubble to get a more accurate estimation of air density. See the new text and recalculated graphical data show below:

Changes to page 20-21

The density of the air inside a bubble was found by using the ideal gas law, $\rho_{air} = P/(R_s T_{air})$, where P is the absolute pressure, $R_s = 287.058 \text{ J/kg}\cdot\text{K}$ is the specific gas constant, and T_{air} is the average temperature inside the bubble which was calculated from the computational results (Fig. S9). These values of T_{air} were used to obtain $\rho_{air} \approx 1.27 \text{ Kg/m}^3$, 1.28 Kg/m^3 , and 1.29 Kg/m^3 for different substrate temperature of $T_w = -20^\circ\text{C}$, -30°C , and -40°C , respectively. Choosing values for p that obtained a best fit to the experimental data results in $p \sim 10 - 100 \mu\text{m}$ (Fig. 6b), consistent with the size of porous features observed within the ice (see Fig. 6c).

Changes to Figure 6

10. What does the symbol [t] mean at the right corner of Fig. 4g?

Thanks to the reviewer for catching glitch. We have fixed it in the revised manuscript.

Changes to Figure 4g

11. page 12, for the simulation (fig. S5), how did the authors model the bubble? Is there any convection effect for the air inside the bubble?

The bubble surface is assumed to be immobile, so natural convection outside the bubble doesn't cause air motion inside of the bubble. We have explained this more clearly in the revised supporting information:

Changes to the section 7 of SI

directly obtained from the built-in material library of COMSOL. Air convection/circulation inside the bubble has not been considered as the bubble surface is assumed to be immobile. In this case, the natural convection outside the bubble does not cause air motion inside the bubble. Moreover, the film thickness ($\sim 1 \mu\text{m}$) is too small to elicit any appreciable thermal resistance across the film. Rather, a sharp interface used and the temperature is assumed continuous across the interface.

12. Fig. 6, is there a case where the liquid film ruptures instead of collapse of the liquid dome?

When using the dish soap as the surfactant, in rare cases the bubble film ruptured before the Laplace-induced collapse could occur. We have mentioned it in the revised manuscript. Moreover, this is shown in a new supporting figure:

Changes to page 21

Np_{avg}^2 where N is the number of pores and p_{avg} is the average pore diameter. For a minority of the room temperature experiments, the liquid dome of the partially frozen bubble ruptured before the Laplace-induced collapse could occur (see Fig. S13). The time scale of bubble rupture (~ 1 ms) is much faster than the collapse event discussed here, further demonstrating that the collapse mechanism is fundamentally different from film rupture.

Changes to the SI

For the new case of pure surfactant (1% SDS), film rupture always occurred far before the dome collapse could occur. This was shown in a new supporting figure that was already shown in response to comment (2).

13. Eq. (11), the numeric factor should be 16 instead of 8?

We thank the reviewer for catching this typographical error; we have fixed it in the revised manuscript. Please note that this typo did not affect the results in the manuscript as we had used the proper numerical factor in the actual simulations.

Changes to page 20

of the pores with a dynamic pressure of $\frac{1}{2}\rho_{air}v_{air}^2$. Following Bernoulli's law, we can equate the dynamic pressure with the Laplace pressure to obtain:

$$v_{air} \approx \sqrt{\frac{16\gamma h}{\rho_{air}(a^2 + h^2)}}. \quad (9)$$

14. In the SI, where is Fig. S11? I could not find it anywhere although it was mentioned in the caption of Fig. S9.

This was a typo in the supporting information. This sentence was referring to what is now Figure 6:

Changes to the caption of Figure S12

substrate (see Fig. S9). In the final frame, the liquid dome atop the bubble collapsed, which is discussed in Figure 6 of the main text.

Reviewer #2:

The authors studied several distinct different icing regimes of the soap bubble by combining scaling analysis, experiments and numerical methods. Two different situations were investigated: For isothermally super-cooled bubbles, ice particles are detached from the freeze front and swirl around the bubble; For the situation when the ambient is warmer than the melting temperature, the freeze front slowly propagates upward and comes to a complete stop at a critical height. In the end of the paper, they used a regime map to summarize the different kinds of freezing behaviors in soap bubbles.

The idea is novel and original, and the results are impressive. The paper was well prepared. I do have some concerns on the manuscript. I would like to see an improved version, in which the issues/comments below have to be satisfactorily addressed.

We thank the reviewer for the encouraging feedback and have responded to these comments below. All changes in the manuscript addressing Reviewer #2 are highlighted in **green**.

Major comments:

1.

- a. Figure 2 and the corresponding discussions: This is a trivial result for film thickness determination. I do not know whether it should be in the main part of the paper. The authors may consider to move it to the supplementary material or the method section.

We agree with the reviewer that the Taylor-Culick approach to film thickness does not contain any new physics. Therefore, we moved the figure to the supporting information. Please also note that most previous uses of the Taylor-Culick law were for planar soap films. One notable case does use it for bubbles (Ref. 32 in revised paper), but does not detail their methodology. Actually, we realized during this revision that our original methodology was inaccurate: we simply tracked the evolving hole diameter without accounting for the translation along the curved surface, which affects the opening velocity. So ironically, it turned out that the correct determination of film thickness was not trivial even to ourselves. Our corrected Fig. S2 and methodology are shown below.

Changes to page 4

hibiting a freezing point of $T_m \approx -6.5^\circ\text{C}$ (see "Methods" section)³⁰. When a calm bubble is punctured, a hole opens and grows due to the unstable surface tension forces at its rim. Using the Dupré-Taylor-Culick law, the initial film thickness of a liquid bubble (e_0) can be determined from the hole-opening velocity by balancing surface tension and inertia^{31,32}:

$$v_b = \sqrt{\frac{2\gamma}{\rho e_0}}, \quad (1)$$

where $\gamma = 24.2 \text{ mN/m}$ is the solution's surface tension measured by using the pendant drop method and waiting until the surfactant had reached a steady-state packing density at the free interface (Fig. S1). Using high-speed imaging, bursting velocities of $v_b \approx 3.2 \text{ m/s}$,

Changes to the SI

Figure S2: (A) Velocity of the hole's rim moving along the arc of the bubble, measured intermittently up until the rim exhibited splashing. Data series correspond to interior volumes of: $\Omega = 5 \mu\text{L}$ (green square), $\Omega = 500 \mu\text{L}$ (red), and $\Omega = 10 \text{ mL}$ (blue). Error bars correspond to one standard deviation between three trials. The air and aluminum substrate were both at room temperature, $T_\infty \approx 22 \pm 1^\circ\text{C}$; the air had a relative humidity of $RH = 26\%$. Constant bursting velocities (solid lines) were found from the average of all the data points for a given bubble size. (B) High-speed image sequences of bursting bubbles used to measure the receding velocities of the liquid rims shown in a. Bubbles were initially punctured with a sharp dry needle at their top. Red arrows in the first row of images show the evolving location of the rim.

- b. Though it is not a novel result, the details of the calculations are still missing. For the Fig 2a, it only records the early stage of the bursting of bubbles. If the authors continue to record the evolution of the L, the scaling will be changed (like the green line). Will be film thickness changed with time?

Please see the revised text pasted above, where these details are now cleared up.

- c. For the Fig 2b, how to calculate the value of L of the bubble when there is a splash at the edge?

We stopped measuring the receding velocity of the liquid rim when the small droplets are ejecting the liquid rim. Please see the revised Fig. S2 pasted above.

- In the Fig 3a, the contact angle of the bubble is changed after being deposited on an ice disk. How to keep the contact angle and the shape of bubbles to be similar on the icy substrate? Will the crystal structure of ice influence the roughness of the substrate and the shape of the bubble? What causes the different areas in the bubble base in the first and second images in Fig 3a?

In the first image of Figure 3a, the top of the bubble is still adhered to the pipette which heavily affects the contact angle on the substrate. After the bubble is completely removed from the pipette tip, its contact angle is approximately constant over time. The spreading time scale (100 us) of the bubble on the icy substrate is much smaller than the freezing time scale (10 s). We have added a supporting figure as well as a sentence to the revised manuscript to clarify this point further.

Changes to the SI

Changes to caption of Figure 2

the release of latent heat. The emissivity coefficient of ice was calibrated at $\epsilon = 0.98$. Time zero corresponds to the bubble's first contact with the icy substrate, where the top of the bubble is still adhered to the pipette (first frames of a–c). d Displacement (δ) of four thermal plumes (different

- Fig 3g: as the tip is pointed, how could the authors define the tip radius of the ice layer?

Please see our response to comment (8) made by Review #1 above.

- Page 7: how to obtain the physical properties of the soap solution, such as the latent heat?

The solution was 80% water by volume, which allows us to approximate the thermophysical properties as pure water for scaling models. Given our model's excellent agreement with the experimental results, we feel that this is reasonable. In two different places in the manuscript,

we have made it more clear why we approximate properties (like density and latent heat) as that of pure water:

Changes to page 4

respectively (Fig. S2). Given that the soap solution is 80% water, we approximate the density (and all other thermophysical properties besides T_m) as that of pure water: $\rho \approx 1000 \text{ kg/m}^3$

5. Page 7, ‘dissipates by thermal conduction across the liquid film...’
What about the convective motion? The authors have to elaborate more on this point.

We have added a new sentence that explains why we expect conduction to dominate convection in the liquid film during solidification:

Changes to page 16

freeze front depending on the direction of the temperature gradient. We neglect convection in the liquid portion of the bubble, as the Marangoni flow has typically dissipated by the time the freeze front has grown appreciably. This balance of heat in versus heat out at the

6. Page 8, the last paragraph ‘Marangoni plumes must be detaching ice ... crack invitation ...’
It does not read clear to me. The authors have to elaborate more on this part.

We have almost completely rewritten the two paragraphs about ice detachment, as seen on the following page:

Changes to page 9-10

Marangoni Freezing and the “Snow-Globe Effect.” Marangoni freezing occurs when Marangoni flows can be generated by freezing-induced heating at the contact line, and these flows dominate over any other possible flow. The two temperature requirements for Marangoni freezing include the condition for freezing: $T_w < T_m$, and the condition for vertically upward Marangoni flow: $dT/dz < 0$, such that $d\gamma/dz > 0$ ³⁸. For the Marangoni flow to dominate, it must be at least as fast as the rate of thermal diffusion, $V_T \sim \alpha_l/\delta$. This first flow criterion is stated in terms of the ratio of the Marangoni velocity, $V_M \sim \Delta\gamma e_0/\eta\delta$ (from Eq. 2), and V_T . This ratio is called the Marangoni number, $Ma = \Delta\gamma e_0/\eta\alpha_l$, which should be greater than or equal to 1. The second flow criterion is that the velocity due to thermal buoyancy, $V_B \sim \Delta\rho g e_0^2/\eta$, must be negligible compared to V_T , resulting in a small Rayleigh number: $Ra \sim \rho g e_0^2 \delta/\alpha_l \eta \ll 1$. The above arguments can be succinctly summarized as $T_t \lesssim T_w < T_m$, $Ma \gtrsim 1$, and $Ra \ll 1$. For the bubbles freezing in the walk-in freezer, all of the conditions were satisfied as $T_t \approx T_w \approx -20^\circ\text{C} < T_m = -6.5^\circ\text{C}$, $Ma \approx 13$, and $Ra \sim 10^{-5}$. Besides generating the thermal plumes, Marangoni freezing can also be responsible for what we call the “Snow-Globe Effect,” as will now be discussed.

Owing to the high Ma number, we propose that Marangoni flows shear off and entrain ice dendrites forming at the bottom-up freeze front. While we do not have any direct evidence of a flow shearing off an ice dendrite, as they are too small to be visible at the point of detachment, there are two strong justifications for this claim. First, it is highly unlikely that hundreds of homogeneous nucleation events would suddenly occur within the liquid film away from the freeze front. This was confirmed by depositing a bubble on a dry silicon wafer (still in the walk-in freezer), where no freezing/nucleation events were observed even after 30 min (Fig. S3c). Second, whenever the suspended ice crystals first appeared (i.e. grew

to a micrometric size), it was always during the Marangoni flow. Indeed, the growth of the entrained ice crystals was often highly asymmetric due to the flow, as seen in Fig. 2b. The Marangoni flow must therefore be detaching invisibly small (i.e. nano-scale) ice particles from the bottom-up freeze front and advecting them upwards. After about 1 s of Marangoni freezing, hundreds of microscopic ice particles were suspended and growing within the film, working in tandem to heat the surrounding liquid. At this point, the gradient in temperature and surface tension is happening in a myriad of locations and directions, as opposed to the original case of a fully out-of-plane gradient extending from the bottom freeze front. Thus

Changes to page 11

This ice detachment can be tentatively modeled by balancing the inertia of the thermal plume (F_i) with the pull-off force required to crack an ice dendrite free of its icy substrate (F_{crack}). For a dendritic contact area of πl^2 , the pull-off force can be determined using the Griffith condition for crack initiation⁴⁰

7. Page 9, ‘Comparing the pull-off force and inertia ($F_i > F_{\text{crack}}$) reveals that a dendrite must be smaller than $l < 10\text{nm}$ for detachment. This is consistent with the observation that detachment events were too small to directly visualize; rather, they were deduced from the flowing crystals later growing to a visible (i.e. micro-scale) size.’ This part is too farfetched! I am not convinced by the estimation of 10 nm, and the authors cannot conclude this number to be corrected because you cannot visualize it. More discussions and revisions have to be provided here. The authors have to be more rigorous.

We agree that the ice detachment model is the most tentative of our models developed here. Therefore, we have added the word “tentative” into our introduction of this model, as shown in the text pasted in point (6) above. We have also extensively rewritten this model as shown in (6) above. Specifically, the emphasis is now on showing all of the experimental evidence that the detachment size is sub-micron, and that this agrees with the model’s tentative prediction of $\sim 10\text{ nm}$. But we no longer emphasize the specific value of 10 nm predicted by the model, we agree that it is more conservative to simply emphasize the sub-micron-scale size more generally. Here we lay out all of the cumulative evidence provided in the paper for the sub-micron size of the detaching ice particles:

- a) Our imaging setup is able to resolve microscopic features, so the fact that we never see the ice particles in the bottom-most portion of the bubble confirms they must be nanoscale at their initial detachment.

- b) We also confirmed that the micro-scale ice particles we can see must be detaching from the bottom freeze front, as ice nucleation never happens in the soap film in the absence of the bottom-up freeze front. Therefore, these crystals are definitely not occurring due to nucleation in the film.
- c) The model's prediction of nanometer-size for detachment is in excellent agreement with our observation of 100 μm particles being visible within 1 s of the start of the freeze front, as the ice is growing at a rate of about 100 μm per second. Therefore, this would predict a negligible (nanoscale) size to the crystals when they first detach, otherwise, they would become visible much closer to the bottom freeze front and in less time.

These arguments are summarized in the following revised text, as shown above in response to comment (6) and also in some more new material shown here:

Changes to pages 11-12:

an ice dendrite and the icy substrate is $w_{\text{ad}} \approx 0.08 \text{ J/m}^2$. The inertia of a thermal plume is $F_i \sim (\rho\pi R_p^2)v_{M,0}^2$, where $v_{M,0}$ is the Marangoni velocity at the very early time limit. Experimentally, we find $v_{M,0} \sim 10 \text{ mm/s}$ by taking the derivative from the $\delta - t$ plot (Fig. 2d) at $t < 10 \text{ ms}$. Balancing the pull-off force and inertia, $F_i \sim F_{\text{crack}}$, predicts that a dendrite must be smaller than $l \lesssim 10 \text{ nm}$ for detachment. As shown in Fig. 2a,b, entrained ice particles grow to $\sim 100 \mu\text{m}$ in size after $\approx 1 \text{ s}$ of bubble deposition on the icy substrate. This is consistent with the measured growth rate of ice of $v_i \sim 100 \mu\text{m/s}$ (Fig. 2e), indicating that ice particles do indeed detach from the freeze front at a nano-scale size.

-
- 8. Page 10, 'This growth rate of $v_i \sim 100 \mu\text{m/s}$ is in agreement with the predicted detachment size of $l < 10\text{nm}$ for dendrites,...' Again, this is too farfetched! What is the direct connection between V_i and the detachment size? This argument is really unconvincing.

We agree that the originally writing was over-reaching; we have made our claims more conservative now as seen in our response to comments (6) and (7) above. It is now made clear that this growth argument can only deduce that the detachment size is sub-micron in general, but cannot validate the specific 10 nm length scale that the model predicts.

9. The heat transfer efficiency between air and liquid is different from that of air and ice. In Fig 4f, have the authors considered this effect?

Regarding the numerical simulations (such as that shown in Figure 4f), we did not consider the fine details of heat transfer across the ice/water film of the bubble. The film thickness is too small to elicit any appreciable thermal resistance across the film. Rather, a sharp interface used and the temperature is assumed continuous across the interface as seen when comparing Figure S8b to S8c. Please see our response to comment (11) made by Review #1 above.

Minor comments:

1. Fig 2 caption: 'curves seen in a' -> 'curves seen in b'.

Thanks to the reviewer for catching this typo. Please see the Fig. S2 in comment (1a) above.

2. Fig 4g: what's the meaning of the [t !] in the bottom right corner?

Thanks to the reviewer for catching glitch. We have fixed it; please see the comment (10) made by Review #1 above.

3. Fig 6 caption: ' ^oC' should be added after 'T_infinite = 25.56'.

This is fixed in the revised manuscript:

Changes to caption of figure 6

Figure 6: Collapse of the liquid dome of partially frozen bubbles. **a** The sudden collapse of the liquid roof of a partially frozen $\Omega = 500 \mu\text{L}$ bubble. Time zero corresponds to the beginning of dome collapse, which was completed in under 1 s. Conditions were $T_w = -20^\circ\text{C}$, $T_\infty = 24.56^\circ\text{C}$, and $RH = 58.8\%$. **b** The height of the liquid dome (h) against time for different substrate temperatures,

4. Fig 6c: Eq. 5 -> Eq. 12.

We have revised the Fig. 6c to address the comment (9) made by Review #1 above.

5. Page 7, 13.5°C -> 13.5 K .

This is referring to a temperature difference, so either is correct and we opted to use Celsius.

Reviewer #3:

Comments:

This work describes experimental and theoretic results on the interesting phenomenon of bubble freezing. Specifically, the authors describe two discrete physical processes that govern the freezing of a bubble in iso-thermal and non-iso-thermal ambient conditions.

The experimental data produced in the manuscript is of high quality and the results are explained clearly. In addition, the scaling analysis and numerical computations augment the ideas derived from the experiments. Overall, the paper is well-written and I think it would of general interest to the readers of the journal. I would recommend the work for publication. However, there are a few comments and questions which authors must address in the revised manuscript.

We thank the reviewer for the encouraging feedback and have responded to these comments below. All the corresponding changes to the manuscript have been highlighted in gray.

A) The abstract begins with a sentence ``\emph{The two-stage freezing process of liquid droplets and films is well known}...'. I am certain that a large number of readers will not be an expert in this field, so a reference to the two-stage freezing process is needed.

Considering that our present paper is not focused on two-stage freezing anyway, we have decided to remove this term entirely and rewrite the first sentence of the abstract:

Changes to the abstract

How droplets and puddles freeze is well known; however, the processes underlying the freezing of bubbles are still a mystery. Herein we investigate the physics

B) page 5: ``\emph{These thermal plumes exhibited a characteristic radius of $R_p \sim 1$ mm}...'. As I understand, a plume is an ascending/descending column of a fluid into another. In that case will it not be sensible to choose the height of the plumes visible in the experimental images as the characteristic length scale rather than R_p ?

We agree that the stand-alone phrase ‘plume’ typically involves one fluid ascending/descending into a separate fluid. However, to our understanding, the phrase ‘thermal plume’ (emphasis on thermal) is specific to temperature alone and does not have to involve mixing of different fluids. Typically, thermal plumes extend from a heat source, which is consistent with our findings of the plumes rising from the bottom-up freeze front.

Regarding the competition of lengths scales, radius versus height, we think both are important. Initially, before the plume has much chance to ascend, it does look somewhat circular in shape which is why we opted to include the effective plume radius. Later on, we agree that the column height becomes important. Regardless, both length scales are much larger than the temperature dissipation length scale, which proves our point that the temperature gradient driving the Marangoni flow is happening entirely within the plumes (as opposed to extending beyond them). We have made this more clear in the revised manuscript:

Changes to page 8-9

A previous work has shown that buoyant thermal plumes can be generated in vertical soap films, where the plumes were primarily inertial³⁶. Inertia is negligible in our system, as the Reynolds number is $Re = \rho V^2 / (\eta V / e_0) = \rho V e_0 / \eta$, where $\eta \approx 2 \times 10^{-3}$ Pa·s is the viscosity of water³⁷ at -6.5°C . For typical values of $V \sim 10$ mm/s, we get $Re \sim 0.01$. A buoyant flow in our soap bubbles would therefore have to balance a gradient in pressure, $\Delta\rho g$, with the gradient in viscous stress, $\eta V / e_0^2$. For a typical value of $\Delta\rho \sim 1$ kg/m³, this leads to buoyant flows of speed $V_B \sim \Delta\rho g e_0^2 / \eta \sim 10$ nm/s. This is in contrast to Fig. 2d, where the speed is not constant over time and is about 6 orders of magnitude faster ($V \sim 10$ mm/s).

This leaves us with the final possibility of a Marangoni flow induced by the latent heat released from freezing. We will refer to this process as “Marangoni freezing.” The freezing-induced heating engenders a gradient in surface tension, $\Delta\gamma / \delta$, where δ is the length scale of the temperature gradient driving the flow. This must be balanced by viscous stress, $\eta V_M / (b + e_0 / 2)$, where V_M is the Marangoni velocity, b is the slip length of the Poiseuille flow along the bubble’s film (Fig. S6), and the velocity profile was approximated as constant-slope. For our system, $b = \sqrt{\eta R / (\rho g t_d)} \sim 1$ μm ⁸, where $t_d \sim 10^3$ s is the drainage time scale of a centimetric bubble which was experimentally observed (Fig. S5a). Therefore $(b + e_0 / 2) \sim e_0$, resulting in a simplified viscous stress of $\eta \dot{\delta} / e_0$, where $\dot{\delta} = d\delta / dt = V_M$ represents the speed a plume. Relating the surface tension stress and viscous stress and solving for δ :

$$\delta \sim \sqrt{\frac{2\Delta\gamma e_0}{\eta}} t^{1/2}. \quad (2)$$

Note that $\Delta\gamma \approx 2$ mN/m for $\Delta T = T_m - T_l \approx 13.5^\circ\text{C}$ corresponding to our degree of supercooling (Fig. 2c)³⁸. When comparing Eq. 2 to experiments, the trajectories of thermal plumes were tracked for $\Omega = 10$ mL bubbles (Fig. 2d). The measurements of δ are in good agreement with 1/2-law with a numerical pre-factor of 1.6, confirming the Marangoni freezing mechanism for flow in the freezing bubbles. Finally, the underlying physics for the resulting wavelength and plume radius ($R_p \sim 1$ mm) are non-trivial and beyond the scope of this present research, as has been noted before in the phenomenon of Marangoni bursting³⁹.

We have also completely updated our writing and several figures regarding aspects of the plumes and the Marangoni flow versus flows due to marginal regeneration, as shown below:

Changes to Figure 2d and its caption

corresponds to the bubble's first contact with the icy substrate, where the top of the bubble is still adhered to the pipette (first frames of a–c). **d** Displacement (δ) of four thermal plumes (different colors) were measured over time when $T_w \approx T_\infty = -19.6^\circ\text{C}$. Inset shows the radius of plumes was of order $R_p \sim 1$ mm. The scale bar represents 2 mm. Time zero corresponds to an estimation of when

Changes to Figure 3 and its caption

Figure 3: Contrasting mechanisms for plumes in non-freezing versus freezing bubbles. **a** For a bubble deposited on a dry, room temperature substrate, plumes were continually generated through the ~ 10 min lifetime of the bubble due to marginal regeneration. **b** For a bubble deposited on an icy cold stage ($T_w = -20^\circ\text{C}$), the bottom-up freeze front (red arrows) suppressed marginal regeneration but enabled a brief (~ 1 s) flow due to Marangoni freezing. In either case here, the ambient conditions were $T_\infty \approx 25^\circ\text{C}$ with a relative humidity of $RH \approx 19\%$.

Changes to Figure S5 and its caption

5 Marginal Regeneration

Figure S5: (A) For a bubble deposited on a dry, room temperature substrate, plumes were continually generated through the ~ 10 min lifetime of the bubble due to marginal regeneration. In this case, the marginal regeneration is likely due to the mismatch in curvatures of the liquid meniscus wetting the surface versus the curvature of the bubble dome. (B) For a bubble deposited on a chilled substrate ($T_w = -20 \pm 1^\circ\text{C}$), marginal regeneration occurs only after the bubble reached to its partially frozen equilibrium (~ 100 s), most likely due to film drainage toward the frozen portion of the bubble. For both (A) and (B), experiments conducted in a room with $T_\infty = 24.2 \pm 0.8^\circ\text{C}$ and $RH = 25 \pm 8\%$. Scale bars represent 5 mm.

Changes to page 18

Marginal regeneration. Once a bubble reached its partially frozen equilibrium, initially there was no appreciable flow in the upper liquid portion of the bubble. After about 100 s, there was a sudden reappearance of plumes within the liquid dome (see Fig. S5b). In contrast to the Marangoni freezing-induced plumes that were observed on initial deposition, these new plumes were because of marginal regeneration. Specifically, the ice-liquid boundary continually thickened at the expense of the top of the liquid dome due to drainage. This was visually evident from the appearance of interference fringes on the thinning liquid dome.

Changes to page 19

The time scale of the formation of these plumes is consistent with the drainage time scale:

$$t_d \sim (\eta R) / (\rho g b^2) \sim 10^2 - 10^3 \text{ s for } R \sim 1-10 \text{ mm}^8.$$

Collapse. After $\mathcal{O}(10 \text{ min})$ of partially frozen equilibrium, the liquid dome suddenly deflated and collapsed (Fig. 6a,b). The time scale from beginning to end of the collapse ranged from $\sim 0.1-10 \text{ s}$, depending on the trial. This gradual deflation of the liquid dome

Changes to page 25

bubble (see Fig. S1A). Plumes (Figs. 3 and S5A) were visualized using a LOWEL DP light which was kept about 5 m away from the experimental set up to minimize heating effects.

C) Caption of figure 3: ``\emph{...due to latent heat.} \rightarrow ...due to the release of latent heat.

We thank the reviewer for this great point. Please note that the previous Fig. 3 is Fig. 2 in the revised manuscript. We have revised the sentence accordingly.

Changes to caption of Figure 2

some of the growing ice particles. c Time-lapse thermographic images, where arrows clarify the bubble-air interface. The liquid portions of the bubble assumed the freezer's temperature shortly after deposition, while the freeze fronts were warmer (i.e. near the melting temperature) due to the release of latent heat. The emissivity coefficient of ice was calibrated at $\epsilon = 0.98$. Time zero

D) page 7: The authors make the statement that ``\emph{... such that Marangoni flow can be induced entirely within a plume}'', is this accurate? Do you expect a strong thermal gradient of 13°C within the plume? A closer look at the thermographic images does not support the statement.

We thank the reviewer for pointing out this apparent inconsistency. The problem was the poor choice of time stamps for Figure 3c. In the first frame ($t = 0 \text{ s}$), the bubble has not had time to cool down to the ambient temperature. But in the second frame ($t = 5 \text{ s}$), there are already so many growing ice crystals within the bubble film that the temperature gradient and Marangoni flow have dissipated. In the revised manuscript, we have chosen new time stamps for this figure so that the temperature gradient driving the initial Marangoni flow is more visible:

Changes to Figure 2c and its caption

E) page 7: The authors use a thermal diffusive length scale δ in the calculation of the viscous drag F_{η} , what is the justification for this particular choice. Instead, the thickness of the bubble e_0 appears to be more appropriate/natural scale.

We have made this point clear in the revised manuscript. Moreover, we have added one supporting figure to describe the terms that we have used in our Marangoni flow.

Changes to the SI

6 Schematic of Flow Inside the Liquid Film

Figure S6: Schematic showing the notations used in Eq. 2 of the main text where the surface tension gradient, $\Delta\gamma/\delta$, was balanced by viscous stress, $\eta V/(b + e_0/2)$. As is shown, δ is the length scale of the temperature gradient driving the flow and b is the slip length of the Poiseuille flow along the bubble's film. Blue arrows show the parabolic velocity profile while the black dashed lines show the constant velocity profile used in this study as an approximation.

F) page 8: Choice of words ``\emph{... the inertia of the Marangoni flow (F_i) ...}'! This terminology is confusing, what do authors want to refer here, inertial or Marangoni forces?

Thanks to the reviewer for catching this point. We meant the inertia of a thermal plume. We made this point clear in the revised manuscript.

Changes to page 11

This ice detachment can be tentatively modeled by balancing the inertia of the thermal plume (F_i) with the pull-off force required to crack an ice dendrite free of its icy substrate (F_{crack}). For a dendritic contact area of πl^2 , the pull-off force can be determined using the Griffith condition for crack initiation⁴⁰

Reviewer #2 (Remarks to the Author):

The authors have addressed all of my concerns in the revised version of the manuscript.

I can recommend the paper to be published in Nature Communications.

Reviewer #3 (Remarks to the Author):

The changes made in the updated manuscript are satisfactory, however, there are still a few minor improvements required:

a) The caption of Fig 2 is confusing: i) The instant $t=0$ is defined twice!?!; ii) Draw a contour over the thermographic images to help the reader identify the interface position, arrows are of practically no use; iii) Crystal growth rates near the substrate and in the bulk are plotted in 1e, does $v \sim 100 \mu\text{m}$ refers to the same?

b) The caption of Fig. S5 is repetitive and confusing. Proofread the supplementary material carefully to weed out such mistakes!

c) Section: Possible Mechanisms \rightarrow I agree with the argument that the internal flow within the bubble at the room temperature is driven by the marginal regeneration mechanism. However, what do the authors mean with the statement: 'Besides, the transient effects in the surface tension measures with the pendant drop were temporal in nature, with no appreciable spatial gradient required to generate flow along the interface'. Can they provide evidence or even a reference to support this statement?

d) Define α_I and T_t immediately after their use!

e) On page 8, the plume velocity $v \sim 10$ mm/s is used without describing how it is calculated! It is defined much later in the manuscript!

f) Further, it is mentioned that the plume velocity is not constant! However, from a glance at the Fig 2d, it appears that the velocity remains constant?

g) Page 10 \rightarrow 'First, it is highly unlikely.....'

I feel authors want to say that the ice crystals nucleation always takes place in the vicinity of the icy substrate and never in the bulk. But how do experiments on a silicon wafer support this argument? In my view since the temperatures in the system never drop below the homogeneous nucleation temperature, thus heterogeneous nucleation happens and it always needs a surface/inhomogeneity in the system to occur.

h) Homogenise the references, see 13, 24, 32

Referee Report for manuscript NCOMMS-18-33887A

Reviewers' Comments:

Reviewer #2:

The authors have addressed all of my concerns in the revised version of the manuscript. I can recommend the paper to be published in Nature Communications.

We thank the reviewer for recommending our manuscript for publication.

Reviewer #3:

The changes made in the updated manuscript are satisfactory, however, there are still a few minor improvements required:

a) The caption of Fig 2 is confusing: i) The instant $t=0$ is defined twice!; ii) Draw a contour over the thermographic images to help the reader identify the interface position, arrows are of practically no use; iii) Crystal growth rates near the substrate and in the bulk are plotted in $1e$, does $v \sim 100 \mu\text{m}$ refers to the same?

We have now removed the accidental repetition of $t=0$ definition. We have also replaced the arrows with dotted-line contours to illustrate the free interface in Fig. 2c. We agree that where the caption referred to the crystal growth rates as an order of magnitude (~ 100) is confusing, we have removed this entirely and simply refer to the approximate values (300 and 250, respectively) of the freeze front speeds.

b) The caption of Fig. S5 is repetitive and confusing. Proofread the supplementary material carefully to weed out such mistakes!

We have read through the Fig. S5 caption several times and do not see where it is repetitive or confusing. Perhaps the reviewer means that some of the caption's information is also mentioned in the main text, but we felt it helpful for the supporting figure caption to stand on its own.

c) Section: Possible Mechanisms \rightarrow I agree with the argument that the internal flow within the bubble at the room temperature is driven by the marginal regeneration mechanism. However, what do the authors mean with the statement: 'Besides, the transient effects in the surface tension measured with the pendant drop were temporal in nature, with no appreciable spatial gradient required to generate flow along the interface'. Can they provide evidence or even a reference to support this statement?

We have rewritten this sentence to make it more clear:

“Besides, the surface tension measured with the pendant drop was only changing temporally, not spatially, as the measured curvature indicated a single value of surface tension for any given time (Supplementary Fig. 1).”

d) Define α_l and T_t immediately after their use!

We thank the reviewer for catching these omissions, they are now defined at first use.

e) On page 8, the plume velocity $V \sim 10$ mm/s is used without describing how it is calculated! It is defined much later in the manuscript!

We now clarify that this velocity is “as measured by observing the initial speed of a rising plume (Fig. 2d)”.

f) Further, it is mentioned that the plume velocity is not constant! However, from a glance at the Fig 2d, it appears that the velocity remains constant!?

The graph in 2d is log-log scale, so the linear slope represents a $\frac{1}{2}$ power law, which is non-linear.

g) Page 10 \rightarrow 'First, it is highly unlikely.....'

I feel authors want to say that the ice crystals nucleation always takes place in the vicinity of the icy substrate and never in the bulk. But how do experiments on a silicon wafer support this argument? In my view since the temperatures in the system never drop below the homogeneous nucleation temperature, thus heterogeneous nucleation happens and it always needs a surface/inhomogeneity in the system to occur.

The argument we gave in the paper is that if the “floating” ice crystals were caused by homogeneous nucleation, then we should see them even for the case of a bubble deposited on a dry silicon substrate without the bottom-up freeze front. The argument the reviewer is giving here is complementary: namely, that the freezer temperature is too high for homogeneous nucleation to occur at all! We have updated the manuscript to include this additional argument; however, our initial argument is still useful for dismissing the possibility of the floating ice because cause by nucleation on solid particulates within the soap film.

h) Homogenise the references, see 13, 24, 32

We have added the page numbers to Ref. 13. We do not see any issues with Refs. 24 or 32 but in general have double-checked all of the references for accuracy.